# The Impact of COVID-19 Crisis upon the Consumer Buying Behavior of Fresh Vegetables Directly from Local Producers. Case Study: The Quarantined Area of Suceava County, Romania

**DOI:** 10.3390/ijerph17155485

**Published:** 2020-07-29

**Authors:** Alina Butu, Ioan Sebastian Brumă, Lucian Tanasă, Steliana Rodino, Codrin Dinu Vasiliu, Sebastian Doboș, Marian Butu

**Affiliations:** 1Biotechnology Department, National Institute of Research and Development for Biological Sciences, 060031 Bucharest, Romania; alina_butu@yahoo.com; 2Romanian Academy, “Gh. Zane” Institute for Economic and Social Research, 700481 Iași, Romania; sebastianbruma1978@gmail.com (I.S.B.); lucian.tanasa@gmail.com (L.T.); codrindinuvasiliu@gmail.com (C.D.V.); sebastian.d.dobos@gmail.com (S.D.); 3Doctoral School Economy II, The Bucharest University of Economic Studies, 010374 Bucharest, Romania

**Keywords:** COVID-19, quarantine, behavior, consumer, short food supply chains, local producers, Romania, Suceava County

## Abstract

The present paper intends to address the impact of COVID-19 crisis upon the consumer buying behavior of fresh vegetables directly from local producers as observed 30 days later, after enforcing the state of emergency in Romania within a well-defined area, namely, the quarantined area of Suceava. The study relies on the interpretation of answers received from the quarantined area (N = 257) to a questionnaire applied online nationwide. The starting point of this paper is the analysis of the sociodemographic factors on the purchasing decision of fresh vegetables directly from local producers before declaring the state of emergency in Romania (16 March 2020). Further research has been conducted by interpreting the changes triggered by the COVID-19 crisis on the purchasing intention of such products before and after the end of the respective crisis. The aim of this scientific investigation relies on identifying the methods by which these behavioral changes can influence the digital transformation of short food supply chains.

## 1. Introduction

Pandemics are not exactly a novel phenomenon strictly related to the current modern societies as they were recorded since ancient times. Each pandemic triggered major changes in economics, regional and global policies, social behavior, and citizens’ mentalities as well. The most significant changes (which have been preserved over the medium term and long term) have been those institutionalized [1]. By contrast, the changes which were least preserved are related to mentalities and social behavior as the institutionalized modifications [2], through public policies, were not sufficiently coupled and consolidated with the psychosocial changes [3]. Like any other pandemics, COVID-19 has caused significant changes on all levels of contemporary society [4,5,6,7,8,9]. All states; continents; regions; urban and rural communities; families; and ultimately, thinking and lifestyle of each individual have been impacted by the pandemic [10,11,12,13,14], and we may never return to the normality previously experienced before COVID-19 [15,16].

At the same time, each pandemic in recorded history had immediate effects on the primary reactions of the social human, because they affected directly health, financial security, life quality and food security [17]. For instance, when cholera or the Spanish flu hit, the economic balance and food supply systems broke and caused famine as an immediate effect [18]. The COVID-19 pandemic has largely fit the same profile, although there are specific differences. This time there have not been major negative effects on food security recorded, except for the underdeveloped and developing countries. Meanwhile, developed economies have not faced serious problems in terms of global food security [19,20]. Inherently, there were individual problems, especially in the case of quarantined people with low and very low income [21]. Nevertheless, the concern for food security has turned into concern for food safety as the public focus of developed countries transitioned to issues such as healthy eating.

However, all pandemics share the demographic vector of disease spread. During the Middle Ages, the pandemic would be transmitted from one part of Europe to another by the people who fled the outbreak [18]. The Amerindian population were decimated by the diseases brought by the European explorers, as they lacked any inherited immunity to the infectious diseases of Europe (in this case, Tzvetan Todorov says that the first globalization was the spread of the viruses [22]). The Spanish flu spread mainly due to the movement of soldiers from the WWI (as they came home back in 1918, they spread the pandemic globally) [23]. The COVID-19 pandemic has been largely triggered by population density [24,25], high degree of mobility of humans, and mass socialization, as well as cultural, social, and tourism events [26,27,28]. Consequently, the measures taken by most world states have addressed issues such as quarantine and isolation, more precisely the enforced social isolation of the population along with the economic isolation between various states or regions as well as between different economic sectors [29]. Hence, this lockdown has impeded the interactions among food systems incorporating every stage of food production and delivery.

If we are to look into Romania’s case, several differences between COVID-19 and the other pandemics should be emphasized. For instance, the Spanish flu, by its direct and indirect effects (which are quite difficult to assess), overlapped the lack of organizational deficiencies of the primary sector, at that time the fundamental branch for achieving the national income (a statistical indicator of the period which is equal to GDP). During the census of 1930, Romania estimated that approximately 50% of material production was provided by the rural population. Therefore, under the circumstances where nearly three quarters of the population were working in agriculture, it is understandable that the Spanish flu had a catastrophic effect on the civil population and army and contributed to the disruption of economic activities. Additionally, during that period, most of Romania’s population had a diet which could be regarded as unbalanced by comparison with the inhabitants of Western Europe. More precisely, monophagous or excessive consumption of cereals had largely contributed to the aggravation of effects of the Spanish flu due to among other factors, a weakened immune system. However in Romania, the COVID-19 pandemic has occurred in a totally different socioeconomic context from the Spanish flu [30,31,32].

On the 14 March 2020, the president of Romania signed the Decree that enforced into law the “state of emergency” for a period of 30 days as of the 16 of March. In this context, the agri-food sector was the first being targeted by the provisions of the Military Ordinance no.1, which drastically limited the activities of public alimentation. Another essential aspect of this military ordinance is granting permission to direct customer delivery in the case of agri-food products. This legal permission provides a favorable context for developing short food supply chains, for Romanian small producers. Additionally, during the COVID-19 pandemic, the number of clients has significantly diminished in hypermarkets and supermarkets, while the convenience stores, butcher shops, and grocery stores have gradually gained more popularity. At the same time, the small producers have seized the opportunity of home delivery for reaching their final customers [33].

Due to increased number of confirmed cases in Suceava and its neighboring localities, the total quarantine was deemed necessary and consequently imposed. To further understand the seriousness of this outbreak, the following clarification is necessary: this area has often been referred to as the “Romanian Lombardy”, a suggestive comparison with the region from northern Italy which was severely affected by COVID-19 [34]. Consequently, on 30 March 2020, the Military ordinance no.6 was adopted regarding implementing total quarantine measures on Suceava municipality and its neighboring localities as shown in Figure 1 [35].

Suceava county covers an area of 8553.5 km^2^, while the quarantined area is 409.4 km^2^. The quarantined area is bordered by Suceava municipality (52.1 km^2^), Salcea town (55.6 km^2^), and the communes of Adâncata (38.6 km^2^), Bosanci (49.6 km^2^), Ipotești (22.8 km^2^), Moara (41.9 km^2^), Șcheia (58.3 km^2^), Pătrăuți (37.7 km^2^), and Mitocu Dragomirnei (52.9 km^2^) [37]. The population of Suceava county was 764,123 inhabitants (3% of Romania’s population) on 1 January 2020. Approximately 25% of Suceava’s population live in the quarantined area, having an irregular territorial distribution (Table 1).

Almost 73% of the quarantined population from Suceava county lives in urban area (67% in Suceava municipality and 6% in Salcea town). The population distribution of age groups, in the quarantined area is as follows: 23% of the population is under 20 years old, 21% is between 20 and 34 years old, 24% ranges between 35 and 49 years old, 19% is between 50 and 64 years old, and 13% is over 65. Therefore, nearly 64% of the population is included in the category of active population (20–64 years old).

Based on the sociocultural characteristics mentioned above, we have worked on the assumption that there are direct correlations among the state of exception induced by quarantine, behavioral changes in the direct purchase of fresh vegetables from producers, and digital transformation of short food supply chains (SFSC). Therefore, the main objective of this research was to identify the possible behavioral changes of the consumers during the COVID-19 crisis, particularly of those customers who bought fresh vegetables with direct delivery. The second objective lies in identifying the possible effects of these behavioral changes on the digital transformation of SFSC.

## 2. Methodology

### 2.1. The Design of Research

A cross-sectional study was carried out based on a survey run between the 10 and 25 April 2020 within the population located in the quarantined area of Suceava (Romania). The basic hypothesis of this paper relies on the assumption that the state of exception that arose from the total quarantine produced significant changes in the consumer buying behavior regarding fresh products directly from local producers. At the same time, these changes can trigger substantial alterations towards the digital transformation of the short food supply chains within the consumer community of Romania.

To test this hypothesis, our research identified sociocultural factors impacting consumer purchasing decisions before the COVID-19 crisis. Further, the analysis has focused on how the quarantine influenced consumer buying behavior and its possible effects in the post-crisis period. The narrative-argumentative sketch of this analysis is outlined in Figure 2.

The survey was based on a questionnaire designed as follows: eight questions meant to identify the sociological profile of the respondents (age, marital status, gender, education, number of people in household, and location); three questions meant to identify the people who bought this type of products before and after 16 March 2020, and those who declared that they would buy after the end of the crisis; one question related to the favorite type of fresh vegetables preferred by respondents; five questions related to the preferred buying method, the frequency of placing orders, the minimum value of order, and payment method.

The questionnaire was pretested between the 8 and 10 April, and all the necessary adjustments revealed during this process were operated by rephrasing several questions and introducing new answer choices. The questionnaire was disseminated through the social media channels and by email. The scope of the survey and objectives of the research were made available to all participants. The participants gave consent and voluntarily joined the study.

The survey was applied one month after the restrictions imposed on free movement of the population were enforced in accordance with the Military Ordinance no. 3. Out of 916 answers received, 257 (28%) are from the quarantined area and represents 0.14% of the total population quarantined, meeting the requirements of the representative sample criterion. The data were compiled, tabulated, and analyzed in accordance with the hypothesis of the study and argumentative-narrative sketch. To better visualize and process the data collected, we have used Excel software (version 365, Microsoft Corporation, Redmond, WA, U.S.A.), open source PSPP software (GNU open source SPSS—Statistical Package for the Social Sciences, Free Software Foundation, Boston, MA, U.S.A.), and R Programming software (version 1.1.456; free software from the R. foundation, R Foundation for Statistical Computing, Vienna, Austria) [39,40,41,42,43,44,45]. Based on this graphic projection, we have run a descriptive analysis which led to anthropological conclusions on the consumer buying behavior and its modifications and impact upon the digital transformation of the short food supply chains as well.

### 2.2. Context

The main concepts employed here are related to short food supply chains (SFSCs), local producers, food safety and security, and sustainable rural development. To have a deeper understanding of the conceptual frame, it is necessary to bring into question the matter of sustainable development, short food supply chains, global and local effects of the crisis upon the SFSCs.

The sustainable rural development in the emerging countries, Romania included, is a necessary condition for these society’s sustainable growth since the population from rural areas still holds a significant percentage in comparison with the more developed economies. The modern development of rural economy involves transition from scale economies to a sustainable local economy, and it is mirrored by the reconfiguration of food supply chain logistics [46,47,48,49,50]. The re-spatialization and reconfiguration of the conventional agricultural systems requires developing new forms of agricultural organization opposed to mass production, namely SFSCs, food hubs, local agri-food systems, and rural networks [51,52,53].

Short food chain supply (SFSC) systems provide multiple benefits (of economic, social, environmental, cultural, and health nature) for people and society as a whole: new job opportunities in the agri-food sector at local level [54], encouraging knowledge transfer, counter-balancing the effects of population migration [55] or gentrification, supporting the local services and suppliers by sustaining the stores of small producers and farmers’ markets, preserving cultural heritage, including promotion of tourism [56] and local gastronomy, and improving the quality of life by securing access to healthier food [57].

At the same time, SFSC systems play a key role in ensuring the quality of the products supplied by direct contact with the producer [58] or by traceability guarantees, although the quality of products is often related to their origin, specifically sustainable agriculture, and particular practices of the post-production process [59]. The fact that SFSCs can facilitate a better interaction between various local agri-food sector producers and potential buyers has been already pointed out by some researchers interested in the development of local food systems. Several authors have highlighted that SFSCs should be considered when discussing how agriculture can be oriented towards more environmentally friendly practices and economic processes while still delivering quality food products [60].

In this context, it should be noted that sometimes short circuits make use of instruments mostly employed by long logistics chains, such as e-commerce, that facilitate the contact between potential buyers and local producer(s) in a certain region. In addition to the effects of demand-side shocks and potential supply-side disruptions, it is worth considering whether the COVID-19 pandemic will have longer-lasting effects on the nature of food supply chains. Two aspects come to mind: (1) the growth of the online grocery delivery sector and (2) the extent to which consumers prioritize “local” food supply chains. An element of food distribution that is undergoing significant change during the COVID-19 pandemic is the expansion of online grocery deliveries [61].

Changing consumption patterns and holding excess inventory in the commercial chains and by customers as well changing the percentage of basic food products/fresh food, collapses in the agro-industrial sector, and exponential increase of online deliveries are impacts from the pandemic in agriculture for most European states, Romania included. Potential challenges include imposing restrictions on the movement of goods, lack of labor force due to the enforced quarantine, closing down important economic agents such as HORECA (Hotels/Restaurants/Café) network, as well as the temporary closure of schools and cafeterias [62]. Regarding goods and services consumed at home, there will be both substitution (positive) and income (negative) effects on demand. The positive substitution effect reflects a switch from out-of-home to in-home consumption (such as the switch from restaurants to home cooking, home delivery, and in-home entertainment) [63]. Another observation is that long food supply chains can be affected by the imposed restrictions through the special measures enforced within the European Union as response to the COVID-19 pandemic. Consequently, the short food supply chains are a complementary solution to the public alimentation, and it can even become a durable and sustainable alternative post crisis.

Given the effects of COVID-19, it is of the utmost importance that countries succeed in keeping the food supply chains running to prevent major food shortages. Accordingly, the Food and Agriculture Organization strongly supports the implementation of concrete measures, mainly specific strategies such as extending emergency food assistance services and offering urgent assistance to smallholders’ agricultural production by improving their involvement in e-commerce [64].

## 3. Analyses and Results

### 3.1. The Descriptive Data on Social Demographics

The primary interest of our narrative sketch was focused on the possible determinations of the sociocultural context on the consumer buying behavior before enforcing the state of emergency. Table 2 summarizes the frequencies of the demographic data of the respondents within this survey.

The majority of survey respondents (85%) live in the urban space of the quarantined area, Suceava—215 (84%) and Salcea—4 (2%). From the rural space of the quarantined area, we received 38 (15%) answers, with the following distribution by locality: Adâncata (3–1.2%), Bosanci (1–0.4%), Ipotești (14–5.4%), Moara (3–1.2%), Șcheia (13–5.1%), Pătrăuți (1–0.4%), and Mitocu Dragomirnei (3–1.2%). Most answers (87%) from all survey respondents (N = 257) come from women. Other prevalent characteristics of respondents were being married, between 20 to 49 years old, having a master’s degree and/or bachelor’s degree, and living with two to four people in their household (Table 2).

### 3.2. Key Factors for the Consumer Buying Behavior

Based on the frequencies observed above, we shall further analyze the key factors for consumer buying behavior. As shown in Figure 3, we generated a biplot based on gender for the multivariate correspondence analysis of the sociodemographic data and purchase of fresh vegetables directly delivered from local producers before the enforcement of the state of emergency (16 March 2020). The validation of the definite data under analysis has been done by using working packages which are typical of R Programming, ggplot2, namely factoextra [39,40,41,42,43,44,45]. In these types of graphs, the four quadrants represent different combinations of sociocultural characteristics we surveyed.

By analyzing the influence of the sociodemographic factors upon the consumer buying behavior of fresh vegetables directly delivered from producers before March 16 (Figure 3), several relevant features can be highlighted and used for outlining the profile of the consumer from the quarantined area. The highest concentration of male gender variables appears in Q1, where there is particular presence of those respondents who did not buy fresh vegetables directly from producers before March 16, and who are included in the age category of 35–49, belong to households of four or five people, and are PhD fellows. The distribution of female respondents is much more uniform and falls nearly equal in all categories of age, education, purchase preference, or persons in the household. The majority of respondents who did not buy fresh vegetables directly from producers before 16 March are located in Q1 and Q4, and are both women and men, especially from the age category of 35–49 years old, of various education levels, and living with four to six people in the same household. Women from the 20–34 and 50–64 age categories, master’s degree holders, and coming from families of two or three people living in the same household appear more inclined to purchasing fresh vegetables directly from producers before March 16.

### 3.3. The Evolution of Buying Behavior

After observing the correlations between the main values in the dataset, we aim to further observe the evolution of consumer buying behavior before, during, and post crisis. In Table 3 the frequencies regarding the purchase of fresh vegetables directly from producers before and after March 16 are presented as well as what is expected to happen after the end of COVID-19 crisis.

The health crisis triggered by COVID-19 has brought major change in purchase behavior towards direct delivery of fresh vegetables from local producers within the quarantined area of Suceava (Figure 4).

Although 88% of the respondents have stated that they did not buy vegetables with direct delivery from producers before the state of emergency was declared, after that date the percentage of consumers who placed direct delivery orders has increased to 60%. Even more, approximately 81% of the respondents have chosen this option post crisis, and only one respondent does not prefer this system of delivery (Table 3 and Figure 4).

It is visible that there is a reversal in the balance of power between those who buy and those who do not buy directly delivered vegetables from producers. It should be noted that in the green-marked columns, there are 31 persons (12%) out of the total of N = 257 respondents who declared that they had bought fresh vegetables directly from producers before March 16, and after this date, their number increased to 154 (60%). The most significant data are to be found in the age groups ranging between 20–34 and 35–49 (Figure 5). Within the age group of 20–34 years old (98 respondents), 13 (5%) people bought before March 16, while 87 (34%) preferred another type of purchase method. After March 16, the number of buyers increased to 67. In the 35–49 age category (123 respondents), 16 (6%) persons bought before March 16, while 107 (42%) favored another type of purchase. After March 16, the number of buyers increased to 72 (29%). Regarding the manner of product selection, the questionnaire data and our research revealed the overwhelming inclination of fresh vegetable consumers to choose specific products and quantities, according to their own needs rather than buying a pre-defined basket.

We are dealing with a localization of vegetable consumption, but in this context, the question arises as to whether it is a psychosocial reaction of the moment in response to the state of emergency, or it is a long-term modification that emerged in the representation system of the consumers from the quarantined area. To have a better understanding of this situation, we shall analyze the distribution of replies depending on the consumers’ statements on what they bought or did not buy before and after March 16 and whether they will keep buying or stop buying after the end of COVID-19 crisis (Figure 6 and Figure 7). The respondents who stated that they bought fresh vegetables from local producers with direct delivery before state of emergency can be regarded as loyal and experienced consumers who prefer this type of purchase, a feature which is still supported, at least in their statements, after overcoming the COVID-19 crisis. Out of 31 respondents who bought fresh vegetables directly from producers before March 16, 29 have stated they will keep buying after the end of COVID-19 crisis (Figure 6). Interestingly, only one respondent of the 31 total respondents expressed no interest in purchasing directly delivered fresh vegetables from local producers after the end of COVID-19 crisis. This person also belongs to the category that did not buy before 16 March 2020.

The imposition of restrictions on free movement, followed by quarantining Suceava municipality and its adjacent localities has caused a fundamental shift among consumers of fresh vegetables who steered towards alternative ways of purchasing produce. After March 16, the number of those who chose to buy fresh vegetables straight from local producers has grown exponentially and reached 154 out of 257 respondents (60%). Among these, 131 (51% of the total percentage) have declared that they will keep ordering vegetables directly from producers post COVID-19 crisis. Among the respondents who did not order vegetables after March 16 (103), 76 (30% of the total percentage) have stated that they will keep buying post COVID-19 crisis (Figure 7).

The emotional state of people living in quarantined areas [65,66,67] leads us to the conclusion that this type of data does not provide enough evidence to consider that after emerging from the COVID-19 crisis, the behavior of direct purchase will increase or have a high inertia. As shown in Figure 8, the abruptly ascending dynamics of the respondents’ number (grouped on clusters related to the purchase intent from before, during, and after the state of emergency) conveys an emotional charge in those who replied to this questionnaire.

However, exposing the customers to this type of experience provides new opportunities for producers and local authorities to support this type of behavior through tailor-made strategies and policies. The purchase frequency can provide additional insight to further analyze this situation. The weekly version of delivery was preferred by respondents (149 answers out of 257) both before and after March 16 (Figure 9). This weekly purchase system has the highest percentage for each age group (57% from the 20–34 age group, 59% from the 35–49 age group, 61% from the 50–64 age group, and 37% from the age group over 65 years old).

At the same time, the weekly purchase is also the logical recommended version, considering the high perishability of the fresh vegetables. Therefore, it is highly likely that buyers will generally adhere to this pattern and choose a weekly purchase. Weekly purchase of produce is the most dynamic and a key vector for enhancing and increasing the trust degree between consumer and producer. The consumer–producer communication has a higher frequency, and the consumer can notice the producers’ compliance with the safety and quality standards over a shorter period.

Another interesting aspect is provided by the similar consumer buying behavior for all age categories considered. The relatively similar distribution of replies presented in Figure 9 shows that all age categories are determined by the same representation system (symbolic systems) related to the preference for the purchase frequency. This situation can be explained by the mainly traditional nature of the quarantined community, the wide access to the same information sources, and the dominantly family nature when purchasing this type of products.

Given the significant number of respondents who have chosen the “Weekly” version, it is obvious that there are solid arguments indicating that post COVID-19 crisis more and more consumers will prefer ordering directly from local producers by weekly home delivery. Cash will remain the favorite payment method. Survey respondents who purchased or plan on purchasing locally before, during, and after the COVID-19 crisis have similar percentage values for frequency of these purchases (Figure 10).

For survey respondents purchasing locally before (31 replies), during (154 replies), and who plan on purchasing after (207 replies) the crisis, preference for weekly orders dropped while preference for ordering every two weeks increased. The structure diagrams previously shown clearly indicate that the percentage of respondents who ordered “when needed” is on the rise, from 10% before the crisis, 17% during the crisis, and 15% post crisis. This shift of behavior, purchasing only when necessary, suggests consumers may be more sensitive to food waste and are aware of the high perishability of fresh vegetables. Accordingly, consumer interest increased for local, fresh, and highly nutritious food, which could impact quality of life for consumers. At the same time, the data obtained reveal a phenomenon of growing confidence in the consumer-local producer relation, and this feature is a key element for mapping and improving the efficiency of the short food supply chains [68].

### 3.4. Key Factors of the Consumer Buying Behavior for the Digital Transformation of the SFSC

The key factors of the consumer buying behavior for the digital transformation of SFSC consist of channels of information and order, frequency of purchasing fresh vegetables, manner of choosing the products, and preferred methods of payment. These factors should be considered by the producers for the digital transformation of their businesses.

#### 3.4.1. Information and Order Channels

Table 4 presents frequencies of online or face-to-face channels preferred by consumers purchasing local produce. As expected, if we take into consideration the social media culture of Romania, Facebook is by far the preferred choice of the respondents regarding the favorite method of information on fresh vegetables distributed through short chains (Figure 11). After Facebook, another top preference is represented by specialized platforms or websites. These three categories of answers can prove useful for both local producers and processors, as well as policymakers with respect to elaborating the digital development strategy of the business or local communities. The Romanian producers who run small- or medium-sized farms are already active on Facebook. However, most of them stopped here since it seemed sufficient for their businesses. They did not develop their own websites nor invested in affiliation with marketing platforms. This type of behavior shows that producers have largely focused on the production activity and ignored diversifying online marketing platforms.

Concurrently, from previous research, we have noticed that the eagerness for growth of small local producers is rather limited. Generally, it is about family or inter-community businesses since expanding the business was regarded as unsustainable. Therefore, before the enforcement of the state of emergency, small producers had mostly focused their businesses on dedicated customer groups. Every producer of the sort developed a network of dedicated clients, a hard-core type of buyers. In this case, two phenomena are likely to emerge. Due to the imposition of the state of emergency and reduced interaction between producers and loyal customers, the dedicated consumers can migrate from one producer to another or to a producer who intends to expand his/her business.

Under the circumstances, digital development will become a survival requirement, not just a mere expansion of the business. In Table 5, the frequencies of the channels preferred by consumers (N = 257) to order fresh vegetables directly from producers can be seen. Similar to the information on channels regarding the fresh vegetables directly delivered from producers, the preferred channels for placing orders are those which proved successful in the case of short food supply chains (Figure 12).

However, based on the data obtained, there are several intriguing behavioral features worth mentioning. Firstly, although phone and email are the tools registering the highest scores of social interaction, they also register the lowest values in the answers received (email: 37 replies out of the total of 257 (14%); phone: 81 answers out of the total number of 257 (31%). This may be explained by the synesthetic nature of the visual consumer’s behavior (here it can be rather related to the desire of seeing the products on a platform, specialized website, or Facebook account). The synesthetic nature is also supported by Romanian consumers symbolically associating agri-food products with childhood memories or idyllic life in the countryside [69].

Secondly, although the online platform and order form are tools which nearly overlap in terms of order technology, the high number of answers received in the category of order form is clearly distinguished here. Again, there is a possible explanation here too. In the case of short food supply chains, in particular fresh vegetables from producers, there are relatively few specialized websites and platforms in Romania. Among existing online platforms, there are even fewer e-commerce apps. This suggests a high degree of wishful thinking by survey respondents. Most answer options are included in the category of online order form (69%, 178 out of 257), online platforms (42%, 107 out of 257), and Facebook (36%, 93 out of 257) (Table 5).

Table 6 presents the frequencies of the preferences (N = 257) for the selection method of fresh vegetables directly from producers. Even under restrictive circumstances, in what concerns the method for selecting products, most respondents (95%) prefer to make their own choices when it comes to products and quantities, while the version based on basket order is favored by merely 5% (Table 6).

This way, the inertia of some specific features is exposed to view by following an empirical line in the case of the purchase decision. The Romanian consumer chooses to control directly and contextually the products and quantities he/she wishes to buy. This piece of information should be noted down and integrated in the distribution strategies on long and medium-term, to achieve an efficient functioning of the short food supply chains. Under the circumstances, the producers should integrate and adjust their offer of products so the consumers could be able to select themselves the ordered products and quantities. The COVID-19 crisis has not changed the buying behavior related to the selection method of fresh vegetables directly delivered by producers. The personal choice of products and quantities is still in the top of most consumers’ preferences over the pre-arranged basket of products as it has been observed throughout all the periods under analysis.

#### 3.4.2. Favorite Methods of Payment

Table 7 introduces the frequencies of the favorite methods of payment for ordering fresh vegetables directly from producers (N = 257). As shown in Table 7, almost half of the respondents (47%) have chosen cash payment, while bank transfer is the least preferred payment method. If we analyze only the answers received from persons choosing home delivery (before, during, and post crisis) the percentage of respondents who use cash payment was lower than for card payments or bank transfers (Figure 13). This demonstrates that during the COVID-19 crisis, the population became aware that card payments or bank transfers can be safe preventive measures against the COVID-19 epidemic. At the same time, this type of behavior is supported by the fact that over this period of crisis, buyers lack cash, since the regulations enforced by the state of emergency have greatly decreased ATM cash withdrawals. To be able to commercialize fresh vegetables with home delivery, local producers need to adjust and facilitate their customers’ payments using Square Point of Sale (POS) or facilitating electronic bank transfers.

We again use multivariate correspondence to analyze the correlations between the sociodemographic data and the consumer buying behavior before March 16, after this date, and post the crisis triggered by the new coronavirus (Figure 14). In the quadrant 1 (Q1) of Figure 14, the strongest correlations are visible among those who ordered fresh vegetables from producers before March 16, those who will order post-crisis, those who order weekly, those who order twice a week, those who prefer to pay cash, and the age category of 35–49. Regarding the order frequency, the highest percentage of respondents who choose direct delivery prefer weekly purchases. Since “After the COVID-19 crisis Yes” is located near the intersection of the two axes from Figure 14 (Q1), this confirms that most respondents will order vegetables with direct delivery after the crisis (207 respondents, 81%). Among these, the highest percentage considers the “Weekly” delivery version as the most convenient one (Q1 in Figure 14). Accordingly, there are relevant arguments indicating that the post COVID-19 period will redirect more and more consumers to weekly home delivery ordering from local producers, while cash remains at the top of preferred payment methods. In Q1, the strongest correlations are between the motivations and purchasing behaviors of high inertia, which favor the purchasing of fresh vegetables directly from producers. The buyer’s profile (accordingly Figure 14) is the following: aged 35–49, purchasing on a weekly basis (once or twice a week), who bought directly from producers before March 16, prefer to pay cash, and will keep buying post-crisis.

Quadrant 2 (Q2) corresponds to those who order fresh vegetables directly from producers monthly, who have not ordered since March 16, are in the 50–64 age category, and prefer to pay by bank transfer. Quadrant 3 (Q3) is associated with people who did not order fresh vegetables directly from producers before March 16, those who will not order or are not certain whether they will order post crisis, and those who prefer ordering every two weeks or whenever it is necessary. They are generally over 65 years old and prefer to pay by debit card. It is highly likely that this profile corresponds to retired people. For them, shopping is a form of socializing, especially in public places. These elderly people normally go shopping for fresh vegetables at farmers’ markets located near their homes. It can be also asserted that this category of consumers does not go shopping to the farmers’ market to support the small local producers (since the go-between traders are present everywhere in these markets, at the expense of small local producers), but they most likely do it by force of habit.

Quadrant 4 is associated with consumers who ordered fresh vegetables directly from producers after March 16 and are in the age category of 20–34. At the same time, the values from Q4 show strong correlations and certain categories from Q1, namely, preference for weekly ordering, payment made by card, and willingness to buy post crisis.

Several aspects of age cluster data distribution warrant further consideration. The clusters of those ranging between 20–34 and 35–49 years old are approximately equally distributed on the horizontal and vertical lines, exhibiting a higher presence in Q1 and Q2. In this area, it is about the high purchasing frequencies and partiality for cash payment. The cluster of those ranging between 50 and 65 years old is mainly distributed to the right side (Q2 and Q3). However, the difference from the distribution visible in Q1 and Q2 is not significant. This suggests the inertia of traditional buying behavior in short food supply chains (SFSCs) is rather higher for older consumers. The cluster of the persons over 65 is mainly distributed in Q3 and shows a high resilience of the purchase of fresh vegetables directly from producers on SFSCs. As we have previously mentioned, such elderly people’s motivations could reside in culture or could be related to socializing needs of this age group. The most representative age categories in the case of the analyzed batch prefers to pay cash and by debit card. Furthermore, those ranging 20–34 years old choose to pay by debit card and cash, while those of 35–49 years old prefer to pay cash and by debit card.

The density of values is represented most closely to the junction of both axes in our multivariate correspondence graphs. Cash or debit card payments show high densities, which are close in terms of percentage. In a Western society, this feature might look slightly concerning, given the strong digital transformation undergone by these cultures. In the case of Romania, the values recorded are still positive and show an increase related to the degree of acceptance of digital transformation in short food supply chains that are currently in an emergent phase in Romania.

## 4. Discussion

There are numerous researchers who have already insisted on the consequences brought by pandemics on the economy, especially on the product distribution systems [70,71,72,73,74,75]. Some of these support the idea that such sanitary crises were followed by economic growth as a direct consequence of the increases in consumption [76,77], while others say that on the contrary, the effects are negative for the human activities [78,79,80], especially for agriculture [81]. When it comes to the current pandemic, Carlsson-Szlezak [82,83] argue there are three types of effects COVID-19 has had on consumption, the market, and distribution chains. This the main reason why we think that the food distribution systems should be redesigned to strengthen resilience in the future to address the complexity of contemporary society [84,85,86,87].

In this context, purchasing fresh vegetables from local producers based on order and direct delivery comes up with a series of advantages, including the fight to reduce the spread of contamination with the new coronavirus. However, consumers avoid shopping at grocery stores, farmers’ markets and/or supermarkets which were often crowded during the pandemic. Social distancing is respected and contact with unknown, possibly infected people is avoided.

To increase local food production and sales in Romania, small local producers need to adhere to shifting customer preferences and innovate their marketing strategies. Cultivation of native varieties, connecting with the local features and measures for environmental protection can constitute advantages for promoting local products in the national agri-food sector. If we take into account the high degree of internet infrastructure development, the e-commerce development of vegetable farmers and virtual farmers’ markets or platforms can also represent solutions for developing businesses in the short food supply chains in times of crises and beyond.

Based on the analyses run in this study, a series of general recommendations stand out for local producers. First, it is advisable for agricultural producers to adjust payment methods to consumers’ demands by purchasing mobile POS systems as well as to develop their own brands and products with an integrated promotion (in analog and digital system). At the same time, and of equal importance, it is imperative for producers to implement up-to-date technologies for placing online orders by developing their own specialized websites and social media. Additionally, innovative marketing and planning of the distribution should be made in accordance with the customers’ demands and short food supply chains. Last but not least, local producers could associate in cooperative organizations for a better access to the market.

For business development, certain limitations need to be taken into consideration. First, supply of fresh vegetables directly from local producers cannot be achieved in Romania throughout the calendar year due to the seasonal nature of crops and reduced areas designated for greenhouses or poly tunnels [88,89]. The quantities and vegetable varieties locally produced and available on the market are reduced due to their zonal and seasonal nature, and consequently, the demand cannot be exclusively covered by local production [88]. Another key factor is the final price of the product, as the price of the local vegetables sold through SFSCs is regularly higher than the prices of similar products in the hypermarket networks [90,91]. To address these issues and come up with viable solutions, the producers could follow the successful associative models from Western Europe thus gaining more visibility and authority in the marketplace. However, the reticence of Romanian small producers is a common denominator when it comes to associating and cooperating locally, including making a common brand, which cannot but become a hindrance for penetrating the local market [92,93,94].

In Romania, the digital transformation of small producers can have a positive effect for the entire economy. However, the digital transformation is also influenced by certain local factors. For instance, to develop distribution channels, small producers need to invest in infrastructure. Nevertheless, their financial possibilities are rather limited, and they choose to invest in means of production at the expense of infrastructure for commercialization and marketing [95]. Furthermore, many small enterprises are in dire need of time, another impediment for digital transformation as they allot most of their activities to production and bringing products to market. Additionally, the scarce digital literacy of agricultural producers is a barrier preventing and limiting the development of the these innovative marketing instruments [96].

Concurrently, we strongly encourage the consumption of fresh vegetables directly purchased from producers and the development of short food supply chains (SFSCs), which bring a series of benefits to the consumers. Typically, local products distributed by SFSCs have superior nutritional value and favorable impacts on people’s general condition and health. Using SPSCs now and into the future can result in indirect economic benefits to consumers by retaining capital locally, which, in turn, has a multiplying effect within the regional economy (maintaining and creating jobs, reinvested profit in productive activities, duties and taxes for the local revenue, etc.) Additionally, the acquisitions made within the SFSCs contribute to environmental protection and hence improve the life quality, especially in urban areas, not to mention the fact that the purchase of fresh vegetables directly from local producers on a regular basis tackles the issue of food waste. Moreover, by consuming local fresh vegetables, the consumer brings his/her own contribution to the preservation of local tradition and identity (local gastronomy, local varieties of vegetables, rural culture, circular rural economy). Finally, the direct delivery of fresh vegetables saves time for consumers by reducing the time spend on purchasing food.

## 5. Conclusions

Our results confirm the hypothesis that the COVID-19 pandemic induced significant changes in consumer purchasing behavior of fresh vegetables. Consequently, consumers are more determined to place online orders of fresh vegetables directly delivered by producers. Prior to enforcing the state of emergency, 12% of the respondents from the quarantined area of Suceava chose the online purchase of fresh vegetables directly delivered by producers. An increased percentage of the respondents (60%) have stated that they intend to adopt this system of buying from short food supply chains (SFSCs) following the COVID-19 crisis.

The preference of consumers for digital instruments of gathering information, ordering, and payment proves that the changes in consumer buying behavior are not merely visible in the purchase intention within this distribution system, but also in their wish for digital transformation of SFCSs. The fact that 95% of the respondents have declared that they prefer a personal selection of the products shows that they choose to involve directly and emotionally in the process of selection and purchase. This is a feature that has not undergone any changes in the timeline determined by the period before March 16, after this date, and after emerging from the COVID-19 crisis.

On the other hand, this study reaches the conclusion that producers should develop their own distribution instruments in a novel manner and by taking into account the preferences shown by the Romanian buyers for high-frequency purchases (weekly or once every two weeks). Thus, SFSCs represent a viable solution to the pandemic, since in Romania’s current context, the reliability and safety of the conventional pattern of agricultural production has been brought into question. Last but not least, to be able to run their businesses under normal circumstances as well, producers should adjust their business philosophy without any delays, in terms of digital transformation, implementation of innovative solutions of distribution addressed to SFSCs, customer communication improvement, and a more appealing presentation of their online product offer [97]. One of the limitations of this scientific investigation is tied to the current preliminary study of just the quarantined area of Suceava. The quarantine of Suceava municipality and its neighboring localities has been the main reason for the geographic limit of our current research. Given the previously mentioned limitations, the results of our study will add to the growing body of research on short food supply chains conducted nationwide in Romania as well as globally.

## Figures and Tables

**Figure 1 ijerph-17-05485-f001:**
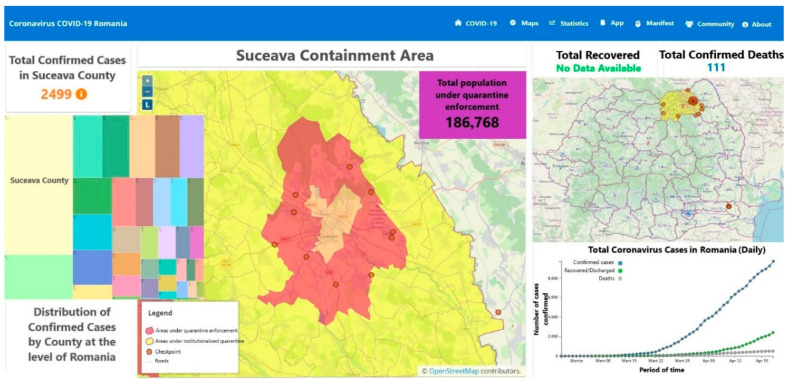
The quarantined area of Suceava. Images processed by authors and reproduced with permission from Geo-spatial.org [36].

**Figure 2 ijerph-17-05485-f002:**
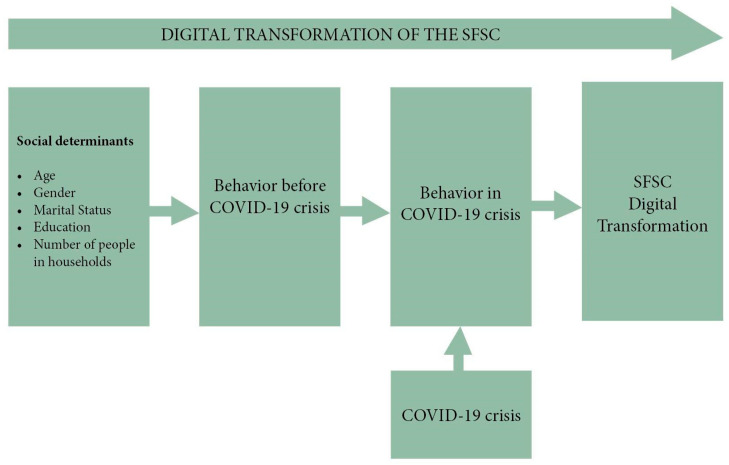
The narrative-argumentative sketch of the proposed research.

**Figure 3 ijerph-17-05485-f003:**
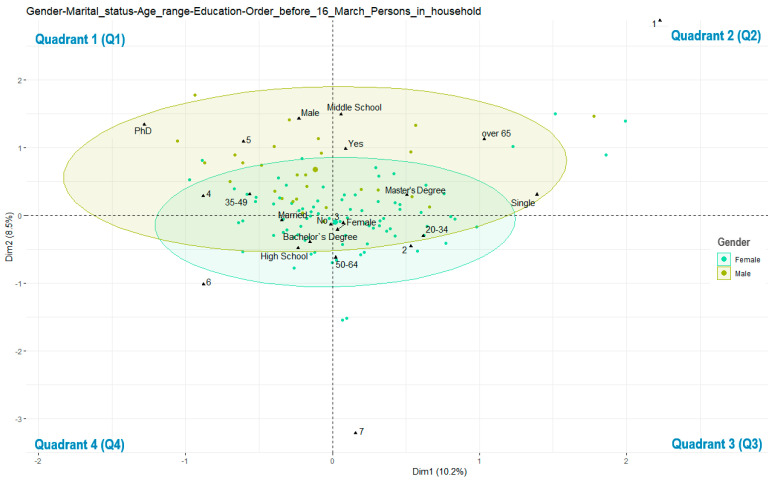
Biplot with multivariate correspondence depending on gender, marital status, age range, persons in household, education, and purchase decision of fresh vegetables directly from producers before March 16.

**Figure 4 ijerph-17-05485-f004:**
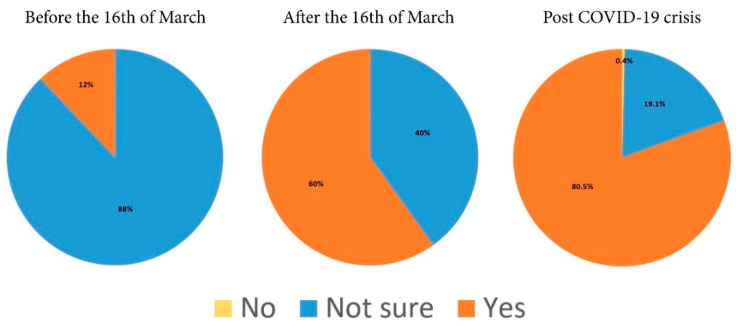
The distribution of answers concerning the purchasing decision of directly delivered fresh vegetables from producers, before and after March 16 and whether they will keep buying post COVID-19 crisis, related to the total number of respondents.

**Figure 5 ijerph-17-05485-f005:**
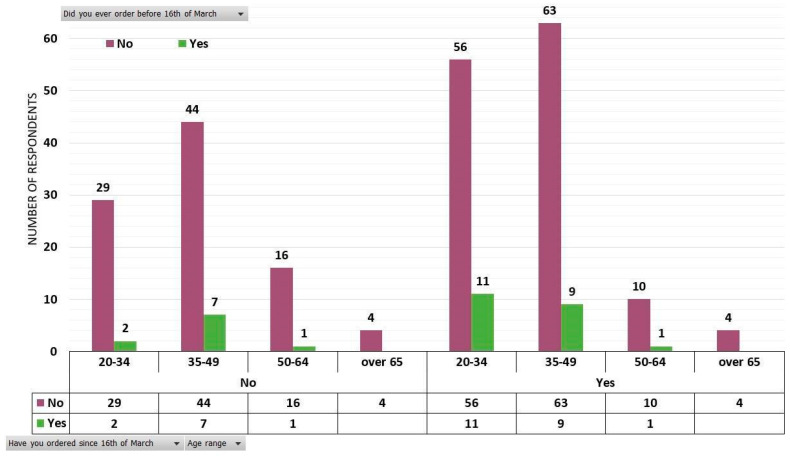
The clusters of those who have declared that they bought or did not buy fresh vegetables from local producers with direct delivery, before March 16, distributed according to age categories and statement that they bought or did not buy directly from producers after March 16.

**Figure 6 ijerph-17-05485-f006:**
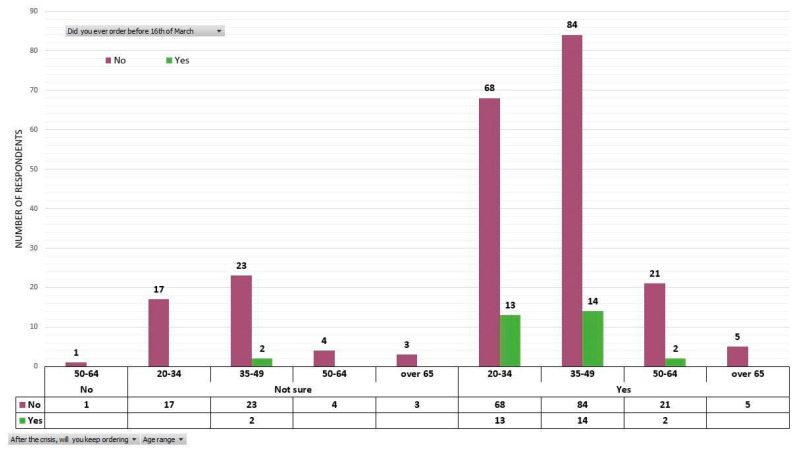
The clusters of those who stated that they bought or did not buy fresh vegetables from local producers with direct delivery before the 16 of March, distributed according to age categories and statement that they will buy or not directly from local producers after the COVID-19 crisis.

**Figure 7 ijerph-17-05485-f007:**
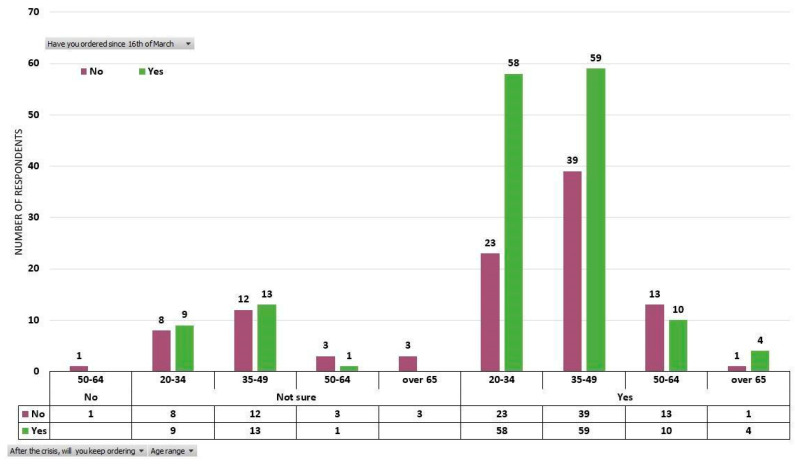
The clusters of those who declared that they bought or did not buy directly delivered fresh vegetables from local producers after March 16, distributed according to age categories and statements that they will keep buying or not from local producers after the COVID-19 crisis.

**Figure 8 ijerph-17-05485-f008:**
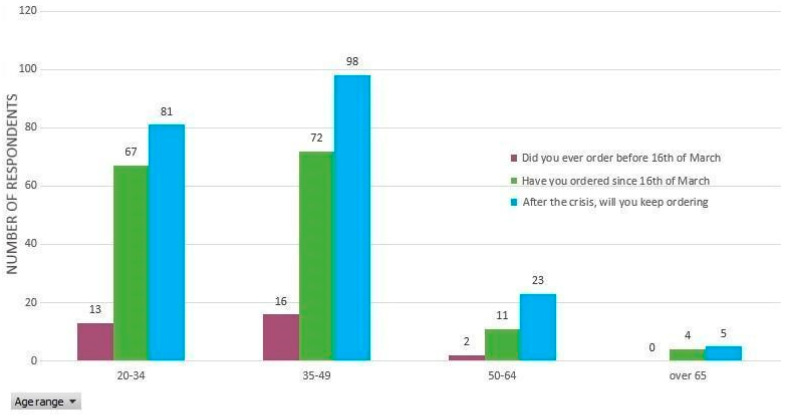
The number of those who declared that they bought directly delivered fresh vegetables straight from local producers before and after March 16 and who will keep buying after the COVID-19 crisis, grouped by age categories.

**Figure 9 ijerph-17-05485-f009:**
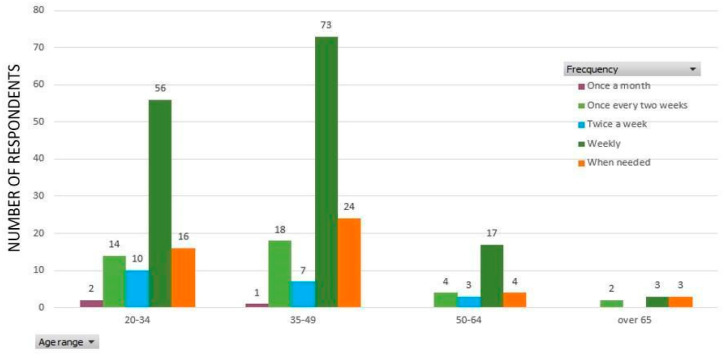
The preferred purchase frequency according to age groups.

**Figure 10 ijerph-17-05485-f010:**
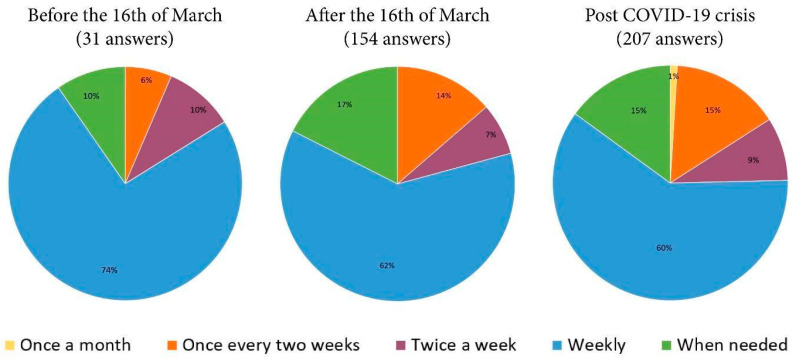
The order frequency on the clusters of the respondents who have stated that they bought fresh vegetables directly from producers before and after March 16 and that they will keep buying post COVID-19 crisis.

**Figure 11 ijerph-17-05485-f011:**
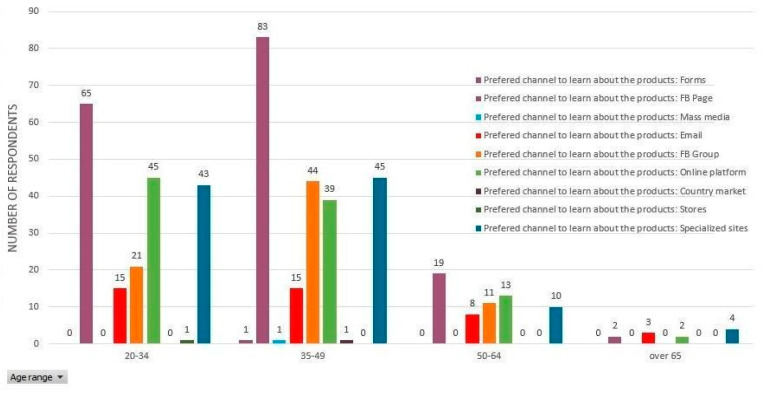
The distribution of answers according to favorite channels of information regarding fresh vegetables, depending on the respondents’ age groups.

**Figure 12 ijerph-17-05485-f012:**
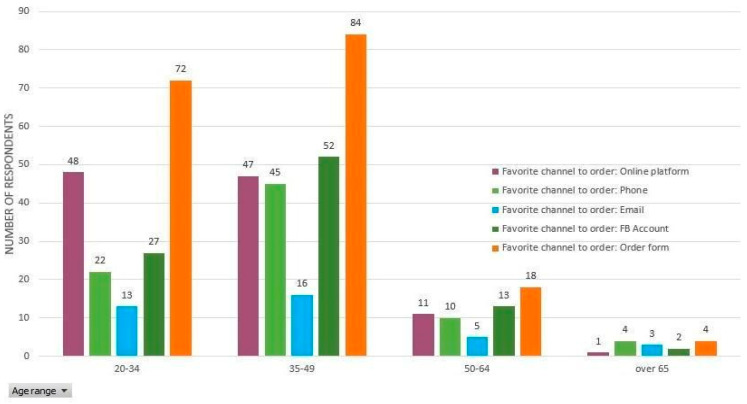
The favorite channels for placing orders depending on age groups.

**Figure 13 ijerph-17-05485-f013:**
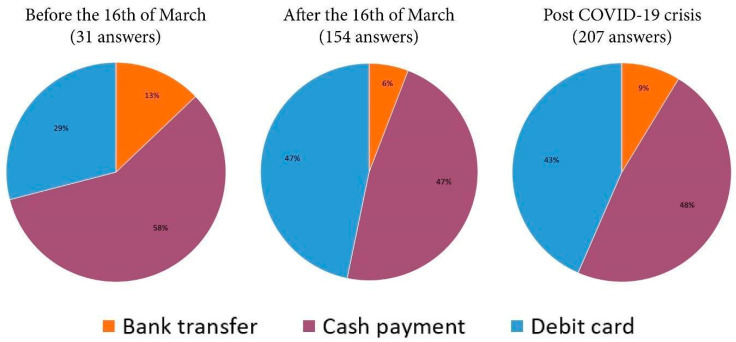
The preferred payment method of the consumers who ordered before and after March 16 and post COVID-19 crisis.

**Figure 14 ijerph-17-05485-f014:**
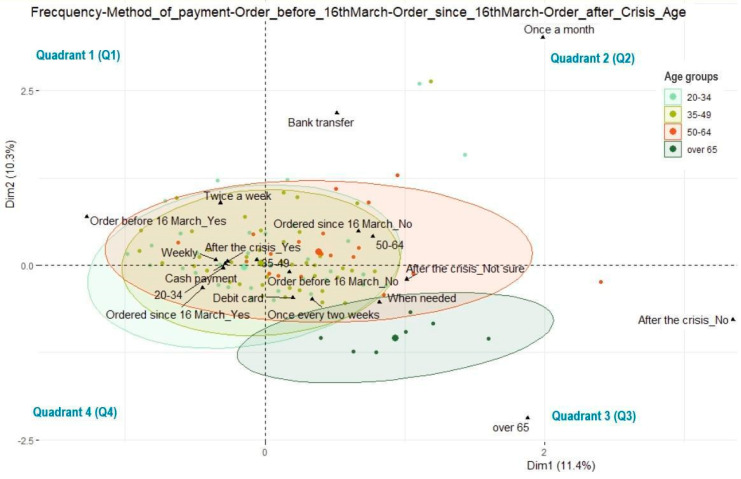
Biplot with multivariate correspondence depending on frequency, age groups, method of payment, and purchase decision of fresh vegetables directly from producers before and after March 16 and post COVID-19 crisis.

**Table 1 ijerph-17-05485-t001:** Population in the quarantined area of Suceava county.

Zone	Total	under 20	20–34	35–49	50–64	over 65
Romania	22,174,693	4,374,106	4,296,857	5,380,505	4,371,494	3,751,731
Suceava County	764,123	179,994	161,690	176,643	134,339	111,457
Quarantine area (total)	186,768	42,214	39,919	45,368	34,953	24,314
Suceava	125,191	24,718	26,730	31,174	25,771	16,798
Salcea	10,931	3250	2339	2413	1512	1417
Adâncata	4359	941	883	1118	714	703
Bosanci	8000	2360	1785	1683	1079	1093
Ipotești	8693	2486	1831	2105	1,85	986
Mitocu Dragomirnei	5489	1641	1270	1163	828	587
Moara	6097	1653	1266	1471	975	732
Pătrăuți	5636	1895	1414	1085	765	477
Șcheia	12,372	3270	2401	3156	2024	1521

Data adapted from NIS database: [38].

**Table 2 ijerph-17-05485-t002:** Frequency distribution table for the social demographics of the respondents.

Variable	Frequency	Valid Percent	Cumulative Percent
Gender			
Female	224	87.16	87.16
Male	32	12.45	99.61
N/A	1	0.39	100
Total	257	100	-
Age Range			
20–34	98	38.13	38.13
35–49	123	47.86	85.99
50–64	28	10.89	96.89
over 65	8	3.11	100
Total	257	100	-
Marital Status			
Married	206	80.16	80.16
Single	51	19.84	100
Total	257	100	-
Education			
Middle School	4	1.56	1.56
High School	28	10.89	12.46
Bachelor’s Degree	118	45.91	58.36
Master’s Degree	90	35.02	93.39
PhD	17	6.61	100
Total	257	100	-
People in household			
1	7	2.72	2.72
2	73	28.4	31.13
3	91	35.41	66.54
4	61	23.74	90.27
5	18	7	97.28
6	5	1.95	99.22
7	2	0.78	100
Total	257	100	-

**Table 3 ijerph-17-05485-t003:** The frequency of those who ordered before 16 March, those who have ordered after 16 March, and of those who will keep ordering after the end of COVID-19 crisis.

Variable	Frequency	Valid Percent	Cumulative Percent
Did you ever order before 16 March			
No	226	87.94	87.94
Yes	31	12.06	100
Total	257	100	-
Have you ordered since 16 March			
No	103	40.08	40.08
Yes	154	59.92	100
Total	257	100	-
After the crisis will you keep ordering			
No	1	0.39	0.39
Not sure	49	19.07	19.46
Yes	207	80.54	100
Total	257	100	-

**Table 4 ijerph-17-05485-t004:** The frequency of the preferred channels for gathering information on the offer of fresh vegetables (N = 257).

Preferred Channel of Information Regarding Fresh Vegetables Offer	Frequency	Valid Percent
Email	41	15.95
Facebook Group	75	29.18
Facebook Page	168	65.37
Online platform	98	38.13
Specialized sites	102	39.69
Forms	1	0.39
Direct from farmers’ market	1	0.39
Direct from producer store	1	0.39
Mass media	1	0.39

**Table 5 ijerph-17-05485-t005:** The frequency of the preferred channels for placing orders (N = 257).

Favorite Channel to Order	Frequency	Valid Percent
Online platform	107	41.63
Phone	81	31.52
Email	37	14.4
Facebook Account	93	36.19
Order form	178	69.26

**Table 6 ijerph-17-05485-t006:** The frequency of the selection method for products (N = 257).

Favorite Way to Order	Frequency	Valid Percent	Cumulative Percent
My choice	245	95.33	95.33
Vendor’s basket	12	4.67	100
Total	257	100	-

**Table 7 ijerph-17-05485-t007:** Frequency of favorite payment methods (N = 257).

Favorite Method of Payment	Frequency	Valid Percent	Cumulative Percent
Cash payment	121	47.08	47.08
Bank transfer	23	8.95	56.03
Debit card	113	43.97	100
Total	257	100	-

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
