# Peer review of "The Impact of COVID-19 Crisis upon the Consumer Buying Behavior of Fresh Vegetables Directly from Local Producers. Case Study: The Quarantined Area of Suceava County, Romania"

_ijerph, 2020, doi:10.3390/ijerph17155485_

Round 1

Reviewer 1 Report

The paper is improved. I thihk is important to publish the paper in this moment considering the subject. 

Author Response

Our response: We'd like to thank you for your message.

Reviewer 2 Report

The overall quality of the paper has significantly improved if compared to previous drafts. The structure of the work now appears clearer and more coherent with the title of the research, although the descriptive statistical methodology remains the main source for the discussion of the results. Although some doubts remain about the representativeness of the sample analyzed, the paper now appears worthy of consideration for the journal. The text, however, requires a complete critical revision by a native english speaker.

Author Response

Our response: We'd like to express our gratitude and appreciation for all of your efforts taken into reviewing our paper as well as for the suggestions provided. We have reviewed once again our manuscript, closely following  all of the suggestions formulated by the reviewers.

Reviewer 3 Report

This manuscript analyzes the impacts of the COVID-19 pandemic on consumer purchasing patterns for vegetables from local farmers in a county in Romania. However, there are substantive short-comings of the manuscript that as a reviewer I cannot recommend publication at this time unless FIVE substantive changes described below and line number specific edits are made. This reviewer would be open to re-evaluating the manuscript after these changes are made:

1) The last paragraph in the introduction is usually one where any hypotheses are stated and then the paragraph ends with a clear statement of objective(s) of the research. The current paragraph states just a hypothesis. Are there any other hypotheses? Sentences need to be added for this as well as a clear statement of the OBJECTIVE(S) of the research.

2) For the quadrant graphs shown in Figure 3 and Figure 14, it needs to be more clearly described in the manuscript what the meaning of each Q1, Q2, Q3, and Q4 quadrat is analytically. Also, please remove the output labeling and edit the axes labels from the default graphical output from R. That would increase clarity.

3) Please add a discussion section between the results section and the conclusion. This should be organized into two parts: 1) the first part should contrast the results from the study with prior literature on impacts on food supply chains from historical disruptions similar to COVID-19 and 2) the second part of the discussion section should focus on key questions and contexts that warrant more discussion based on the results. For example, the results suggest that a shift to online ordering presents an excellent opportunity for local farmers to increase their market share and stability of their customer base. However, this may not be possible if broadband internet access is limited in the areas these farmers are located. There may be other challenges to shifting to this new marketing domain such as the opportunity cost of time to take orders and deliver to individuals (increases transaction costs), rather than having all the individuals come to you the farmer (e.g. farmers market, CSA pick-up, farm stand, etc.). Increased time spent taking orders and delivering (e.g. economies of scope) could be used for expansion and reaching economies of scale. While this does not exhaust potential limitations to this new way of marketing, time should be taken by the co-authors of this manuscript to listing what these are and writing about how to address these limitations. Another potential discussion topic could be clarifying if this new consumer base for local farmers is as permanent as one thinks this could be. Could the industrial food system’s supermarkets and retailers also shift to online ordering and parking lot pick-up or delivery and eventually negate the gains made by local farmers? Other discussion topics would be great if the ones suggested are not pertinent.

4) The manuscript needs to be edited for English, especially improvements of flow and understanding. For example at the start of the last paragraph in the introduction: “Having as a starting point the features mentioned above, the authors of this scientific investigation work on the assumption” can be replace with just writing directly about the point being made. The excessive use of wording that does not contribute to the primary point makes understanding the writing extremely difficult. “Having as a starting point” anywhere in the manuscript should not be used. Just go straight to the main point of the sentence. Also, paragraphs need to have a minimum of three sentences. Single sentence paragraphs need to be consolidated into other paragraphs.

5) Fourteen figures is a lot for a manuscript. Please create a Supplemental Materials section for some of these figures which can be referenced in the manuscript (e.g. Figure S1) without having to take up so much of the body of the manuscript (the figures that make the most sense are the ones getting into more detail regarding variation in the response by age class):

  1. Figure 5 – Change to Figure S1 in Supplemental Materials.
  2. Figure 6 – Change to Figure S2 in Supplemental Materials.
  3. Figure 7 – Change to Figure S3 in Supplemental Materials.
  4. Figure 8 – Change to Figure S4 in Supplemental Materials.
  5. Figure 9 – Change to Figure S5 in Supplemental Materials.
  6. Figure 11 – Change to Figure S6 in Supplemental Materials.
  7. Figure 12 - Change to Figure S7 in Supplemental Materials.
  8. Create a Supplemental Materials Table S1 which would be a table of the complete survey used. Make sure to reference this as Table S1 in the manuscript text.

So Figures 1 through 4 remain unchanged in the manuscript. Figure 10 changes to Figure 5, Figure 13 changes to Figure 6, and Figure 14 changes to Figure 7. That would make 7 Tables and only 7 Figures which would increase clarity.

Specific Line Number of Manuscript edits (note that requested change of word(s) in quotations):

L32 – Remove “effect of this”

L51 – Change to “from northern Italy”

L54 – Change to “as shown in Figure 1”

L55 (Figure 1) – Change to “186,768” and use comma and not period for thousands place.

L58 – Remove “of”

L62-64 – “The population of Suceava country was 764,123 people (3.44% of Romania’s population) on 1st January 2020. Approximately 25% (186,768 people) lived in the quarantined area,”

L64 – Change to “(Table 1)”

L67 – Change to “lives in urban areas”

L95-102 – Do not write using bullet points but rather write in paragraph form.

L99 – Change to “2020”

L133 – Change to “opposed to mass production”

L136 – Change to “health for”

L137 – Change to “food sector at the local level [25], encouraging knowledge transfer,”

L141 – Change to “to healthier food”

L155-157 – Change to “Two aspects come to mind: 1) the growth of the online delivery sector and 2) the extent to which consumers prioritize “local” food supply chains.”

L170 – Change to “in response”

L182 – Change to “Table 2 summarizes frequencies”

L184 (Table 2) – Change header to “Cumulative Percent” and if there is no difference in the percent column and the valid percent column, please use just ONE column as percent. Also switch the PhD row with the Post Graduate Degree row and rename “Post Graduate Degree” to “Master’s Degree” if that is indeed what a “Post Graduate Degree” is (degree after college B.S. or B.A.)

L191-204 – Re-write this into ONE paragraph and write in a way so the writing has less numbers and is more descriptive.

L206 – Make sure the indent to this paragraph is consistent with the other paragraphs.

L220-231 – Re-write this into ONE paragraph and write in a way so the writing has less numbers and is more descriptive.

L233-235 – Re-write to increase clarity.

L238 (Table 3) – Change header to “Cumulative Percent” and if there is no difference in the percent column and the valid percent column, please use just ONE column as percent.

L240-242 – Re-write to increase clarity.

L253-266 – Re-write this into ONE paragraph and write in a way so the writing has less numbers and is more descriptive.

L284-286 – Re-write to increase clarity and add to the end of the preceding paragraph.

L314-323 – Re-write this into ONE paragraph and write in a way so the writing has less numbers and is more descriptive.

L326-348 – Re-write so organized into longer paragraphs.

L379-380 (Table 4) – Change header to “Cumulative Percent” and if there is no difference in the percent column and the valid percent column, please use just ONE column as percent.

L408 (Table 5) – Change header to “Cumulative Percent” and if there is no difference in the percent column and the valid percent column, please use just ONE column as percent.

L437 (Table 6) – Change header to “Cumulative Percent” and if there is no difference in the percent column and the valid percent column, please use just ONE column as percent.

L457 (Table 7) – Change header to “Cumulative Percent” and if there is no difference in the percent column and the valid percent column, please use just ONE column as percent.

L515-531 – Do not write using bullet points but rather write in paragraph form.

L565-570 – Get rid of this manuscript section and merge into the end of the conclusions section above.

Author Response

This manuscript analyzes the impacts of the COVID-19 pandemic on consumer purchasing patterns for vegetables from local farmers in a county in Romania. However, there are substantive short-comings of the manuscript that as a reviewer I cannot recommend publication at this time unless FIVE substantive changes described below and line number specific edits are made. This reviewer would be open to re-evaluating the manuscript after these changes are made:

1) The last paragraph in the introduction is usually one where any hypotheses are stated and then the paragraph ends with a clear statement of objective(s) of the research. The current paragraph states just a hypothesis. Are there any other hypotheses? Sentences need to be added for this as well as a clear statement of the OBJECTIVE(S) of the research.

Our response: We have revised the manuscript strictly according to your recommendations and suggestions. More precisely, at the end of the “Introduction” section we have stressed better the objectives of our study.

2) For the quadrant graphs shown in Figure 3 and Figure 14, it needs to be more clearly described in the manuscript what the meaning of each Q1, Q2, Q3, and Q4 quadrat is analytically. Also, please remove the output labeling and edit the axes labels from the default graphical output from R. That would increase clarity.

Our response: Given our choice of rendering we can only share with you the fact that these quadrants are used in RProgramming simply for the purpose of having a better overview on the issue. The same applies for the axis. Moreover, we have to simply admit that the standard approach of the analysis was the one we really sought after. One can observe the correlations in each quadrant and then can see the overall picture when pondering all of the four areas. The more the variables are rendered near the XOY center the higher the density and higher number of respondents. When considering the angles of variables rendered in relation to the OY axis, one can see that the lower the angle the higher the correlation intensity. The higher the angle degree basically points out to a lower intensity correlation. We considered that by labeling we`d hamper the insight and clarity of a PCA (Principal Component Analysis) biplot. However, when addressing the issue of a series of variables, indeed labelling is of utmost importance.

3) Please add a discussion section between the results section and the conclusion. This should be organized into two parts:

3.1) the first part should contrast the results from the study with prior literature on impacts on food supply chains from historical disruptions similar to COVID-19 and

Our response: Given the topic of our draft paper we can only assure you that, in recent times. Romania has not been faced with anything similar to the COVID-19 healthcare crisis. As a result, there are no works available on the matter in Romanian literature thus far. We simply had no reference articles to relate to or follow up as guidance a.s.o. at hand given our topic of choice.

3.2) the second part of the discussion section should focus on key questions and contexts that warrant more discussion based on the results. For example, the results suggest that a shift to online ordering presents an excellent opportunity for local farmers to increase their market share and stability of their customer base. However, this may not be possible if broadband internet access is limited in the areas these farmers are located. There may be other challenges to shifting to this new marketing domain such as the opportunity cost of time to take orders and deliver to individuals (increases transaction costs), rather than having all the individuals come to you the farmer (e.g. farmers market, CSA pick-up, farm stand, etc.). Increased time spent taking orders and delivering (e.g. economies of scope) could be used for expansion and reaching economies of scale. While this does not exhaust potential limitations to this new way of marketing, time should be taken by the co-authors of this manuscript to listing what these are and writing about how to address these limitations. Another potential discussion topic could be clarifying if this new consumer base for local farmers is as permanent as one thinks this could be. Could the industrial food system’s supermarkets and retailers also shift to online ordering and parking lot pick-up or delivery and eventually negate the gains made by local farmers? Other discussion topics would be great if the ones suggested are not pertinent.

Our response: We have made all the necessary efforts aiming at reviewing our manuscript/draft paper according to your suggestions and highly relevant recommendations. We fully agree with your view on the matter as well as issues pointed out. We formulated the necessary add-ons.

4) The manuscript needs to be edited for English, especially improvements of flow and understanding. For example at the start of the last paragraph in the introduction: “Having as a starting point the features mentioned above, the authors of this scientific investigation work on the assumption” can be replace with just writing directly about the point being made. The excessive use of wording that does not contribute to the primary point makes understanding the writing extremely difficult. “Having as a starting point” anywhere in the manuscript should not be used. Just go straight to the main point of the sentence. Also, paragraphs need to have a minimum of three sentences. Single sentence paragraphs need to be consolidated into other paragraphs.

Our response: We had solved it.

5) Fourteen figures is a lot for a manuscript. Please create a Supplemental Materials section for some of these figures which can be referenced in the manuscript (e.g. Figure S1) without having to take up so much of the body of the manuscript (the figures that make the most sense are the ones getting into more detail regarding variation in the response by age class):

  1. Figure 5 – Change to Figure S1 in Supplemental Materials.
  2. Figure 6 – Change to Figure S2 in Supplemental Materials.
  3. Figure 7 – Change to Figure S3 in Supplemental Materials.
  4. Figure 8 – Change to Figure S4 in Supplemental Materials.
  5. Figure 9 – Change to Figure S5 in Supplemental Materials.
  6. Figure 11 – Change to Figure S6 in Supplemental Materials.
  7. Figure 12 - Change to Figure S7 in Supplemental Materials.
  8. Create a Supplemental Materials Table S1 which would be a table of the complete survey used. Make sure to reference this as Table S1 in the manuscript text.

So Figures 1 through 4 remain unchanged in the manuscript. Figure 10 changes to Figure 5, Figure 13 changes to Figure 6, and Figure 14 changes to Figure 7. That would make 7 Tables and only 7 Figures which would increase clarity.

Our response: We think that we don't have any reasonable arguments to say that your point is not correct. At the beginning, we`ve given it quite some time of thinking. While pondering the length of the section we somehow reached the conclusion that maybe it was for the best for the reader to have say more figures at hand given our topic. We were under the impression that our approach would have better suited the reader. In any case, we have revised our paper according to your suggestions.  We can only share with you the fact that the table entitled complete survey used was and still is the very database used during our data processing. Please find attached our questionnaire deployed during all of our research as “supplementary material”, according to your recommendation.

Specific Line Number of Manuscript edits (note that requested change of word(s) in quotations):

L32 – Remove “effect of this”

Our response: We had solved it.

L51 – Change to “from northern Italy”

Our response: We had solved it.

L54 – Change to “as shown in Figure 1”

Our response: We had solved it.

L55 (Figure 1) – Change to “186,768” and use comma and not period for thousands place.

Our response: We had solved it.

L58 – Remove “of”

Our response: We had solved it.

L62-64 – “The population of Suceava country was 764,123 people (3.44% of Romania’s population) on 1st January 2020. Approximately 25% (186,768 people) lived in the quarantined area,”

Our response: We had solved it.

L64 – Change to “(Table 1)”

Our response: We had solved it.

L67 – Change to “lives in urban areas”

Our response: We had solved it.

L95-102 – Do not write using bullet points but rather write in paragraph form.

Our response: We had solved it.

L99 – Change to “2020”

Our response: We had solved it.

L133 – Change to “opposed to mass production”

Our response: We had solved it.

L136 – Change to “health for”

Our response: We had solved it.

L137 – Change to “food sector at the local level [25], encouraging knowledge transfer,”

Our response: We had solved it.

L141 – Change to “to healthier food”

Our response: We had solved it.

L155-157 – Change to “Two aspects come to mind: 1) the growth of the online delivery sector and 2) the extent to which consumers prioritize “local” food supply chains.”

Our response: We had solved it.

L170 – Change to “in response”

Our response: We had solved it.

L182 – Change to “Table 2 summarizes frequencies”

Our response: We had solved it.

L184 (Table 2) – Change header to “Cumulative Percent” and if there is no difference in the percent column and the valid percent column, please use just ONE column as percent. Also switch the PhD row with the Post Graduate Degree row and rename “Post Graduate Degree” to “Master’s Degree” if that is indeed what a “Post Graduate Degree” is (degree after college B.S. or B.A.)

Our response: We had solved it.

L191-204 – Re-write this into ONE paragraph and write in a way so the writing has less numbers and is more descriptive.

Our response: We had solved it.

L206 – Make sure the indent to this paragraph is consistent with the other paragraphs.

Our response: We had solved it.

L220-231 – Re-write this into ONE paragraph and write in a way so the writing has less numbers and is more descriptive.

Our response: We had solved it.

L233-235 – Re-write to increase clarity.

Our response: We had solved it.

L238 (Table 3) – Change header to “Cumulative Percent” and if there is no difference in the percent column and the valid percent column, please use just ONE column as percent.

Our response: We had solved it.

L240-242 – Re-write to increase clarity.

Our response: We had solved it.

L253-266 – Re-write this into ONE paragraph and write in a way so the writing has less numbers and is more descriptive.

Our response: We had solved it.

L284-286 – Re-write to increase clarity and add to the end of the preceding paragraph.

Our response: We had solved it.

L314-323 – Re-write this into ONE paragraph and write in a way so the writing has less numbers and is more descriptive.

Our response: We had solved it.

L326-348 – Re-write so organized into longer paragraphs.

Our response: We had solved it.

L379-380 (Table 4) – Change header to “Cumulative Percent” and if there is no difference in the percent column and the valid percent column, please use just ONE column as percent.

Our response: We had solved it.

L408 (Table 5) – Change header to “Cumulative Percent” and if there is no difference in the percent column and the valid percent column, please use just ONE column as percent.

Our response: We had solved it.

L437 (Table 6) – Change header to “Cumulative Percent” and if there is no difference in the percent column and the valid percent column, please use just ONE column as percent.

Our response: We had solved it.

L457 (Table 7) – Change header to “Cumulative Percent” and if there is no difference in the percent column and the valid percent column, please use just ONE column as percent.

Our response: We had solved it.

L515-531 – Do not write using bullet points but rather write in paragraph form.

Our response: We had solved it.

L565-570 – Get rid of this manuscript section and merge into the end of the conclusions section above.

Our response: We had solved it.

Reviewer 4 Report

This is one of the most powerful and necessary papers I have read on the covid-19 topic. It deserves publication and attention, as it draws on locally implemented surveys and brings into foreground important insights into the domestic food chain supplies and their relationship to consumers, based on international criteria and standards of analytical approach. I believe it delivers vital information on ways to improve the local infra-structures of delivery consumption as well as a valid model for expanding this type of  research on a broader - even national - area, with possible extension for other areas of the Romanian industry and economy, such as HoReCa (tourism), the artistic department (cultural tourism) or education.

Author Response

This is one of the most powerful and necessary papers I have read on the covid-19 topic. It deserves publication and attention, as it draws on locally implemented surveys and brings into foreground important insights into the domestic food chain supplies and their relationship to consumers, based on international criteria and standards of analytical approach. I believe it delivers vital information on ways to improve the local infra-structures of delivery consumption as well as a valid model for expanding this type of  research on a broader - even national - area, with possible extension for other areas of the Romanian industry and economy, such as HoReCa (tourism), the artistic department (cultural tourism) or education.

Our response: We honestly value your appreciation of our efforts as well as the time taken for reviewing our draft paper. We feel but compelled to consider it as a motivation given that it is not easy at all to follow this particularly difficult topic. However, we've tried our best to follow up your recommendations and useful suggestions. Your insight is of utmost relevance to our research efforts.

Line Remark

30 Add “the contemporary society”

Our response: Done

31 Replace “finally” with “ultimately”

Our response: Done

41 Delete “the” from “the Romanian small producers”

Our response: Done

99 Change “20202” into “2020”

Our response: Done

126 Replace “global and local effects” with “global as well as local effects”

Our response: Done

163 Add “the” in front of HORECA

Our response: Done

304 Delete the comma after “[36-38]”

Our response: Done

421 Replace “or” with “of”

Our response: Done

541 Add “the” in “by COVID-19 crisis”

Our response: Done

Round 2

Reviewer 3 Report

This manuscript analyzes the impacts of the COVID-19 pandemic on consumer purchasing patterns for vegetables from local farmers in a county in Romania. However, there are substantive short-comings of the manuscript that as a reviewer I cannot recommend publication at this time unless FOUR substantive changes described below and line number specific edits are made. The first substantive change was not addressed from the initial review. This reviewer would be open to re-evaluating the manuscript after these changes are made:

1) The first part of the discussion should contrast the results from the study with prior literature on impacts on food supply chains from historical disruptions similar to COVID-19 (e.g. 1918-19 Spanish flu, etc.). This will also increase the number of citations as the number currently is low. This was requested during the first review but not done. The research needs to show how it fits into the broader literature. Since COVID-19 is currently happening, the contrasts of the research results with other literature are likely to be literature on similar disruptions to food marketing from other historical events in Romania and elsewhere in the world.

2) The last part of the discussion needs to add a paragraph at the end on how to address the limits agricultural producers and consumers face when it comes to following these recommendations that have been made in the prior two paragraphs. That is really at the heart of the discussion so if this is more than one paragraph that is no problem. The end of the currently written discussion offers recommendations based on the results. That is great but the discussion needs to go beyond this and discuss the implications on how practical these recommendations are and how any barriers and challenges to following these recommendations can be substantively addressed.

3) After conducting more literature search on this, there may be enough information to add a paragraph to the introduction on historical impacts to agricultural marketing during prior pandemics or disruptions to introduce the reader to the context of the analysis.

4) What is the difference between 12.06% and 12%? Rounding off does not change the point that is being made. Including that many decimal places actually increases the difficulty of interpreting results. Please round off to the nearest percent throughout the written manuscript.

Specific Line Number of Manuscript edits (note that requested change of word(s) in quotations) and places denoted by “???” where the co-authors will have to write in to clarify:

L30 – Remove “the”

L51 – Remove “it is”

L56-62 – These are two paragraphs…merge into ONE paragraph

L66 – Change to “The population distribution of age groups,”

L71 – Change to “Based on the sociocultural characteristics mentioned above,”

L73 – Change to “short food supply chains (SFSC).”

L75 – Change to “during the COVID-19 crisis,”

L80-86 – These are two paragraphs…merge into ONE paragraph

L87-88 – Change to “To test this hypothesis, our research identified sociocultural factors impacting consumer purchasing decisions before the COVID-19 crisis.”

L89 – Change to “how the quarantine influenced consumer buying behavior…”

L90 – Change to “The narrative-argumentative sketch of this analysis is outlined in Figure 2.”

L91-92 – Reduce the size of Figure 2 so it fits in the blank part of the previous page (you will have to do this of page proof edits)

L94-100 – To be consistent with other parts of the manuscript, please use numbers and write out number of questions…for example, change to “as follows: 1) eight questions…” and “location), 2) three questions…” and “the crisis, 3) one question…” and “by respondents, and 4) five questions…”

L104 – Change to “…were made available...”

L106-115 – These are three paragraphs…merge into ONE paragraph

L115 – Change to “of short food supply chains as well” and

L117 – Change to “…related to short food supply chains (SFSCs), local producers, food”

L120 – Change to “…upon the SFSCs.”

L127 – Change to “…namely SFSCs, food hubs, local agrifood”

L129 – Change to “Short food chain supply (SFSC) systems…”

L130 – Change to “health nature) for people and society as a whole: new job…”

L133 – Change to “…preserving cultural heritage, including…”

L134 – Change to “…securing access to healthier”

L136 – Change to “At the same time, SFSC systems…”

L146-147 – Change to “…between potential buyers and local producer(s) in a certain region.”

L153-158 – Change to “Changing consumption patterns, holding…” Also, break up this very long sentence into two separate sentences to improve the reader’s ability to process this so change “…fresh food, ellipses in the agro-industrial sector, and exponential increase of online deliveries are impacts from the pandemic in agriculture for most European states, Romania included. Potential challenges include imposing restrictions on the movement of goods, lack of labor force due to the enforced quarantine, closing down important economic agents such as HORECA network, as well as the temporary closure of schools and cafeterias [33].”

L157 – You need to state what this HORECA stands for so change to “such as ??? (HORECA) network, as well…”

L175-191 – These are three paragraphs…merge into ONE paragraph and if you are putting Table 2 between the paragraph then do NOT indent the second part.

L179 – Also, please do not start a sentence with a numerical percentage so change to “The majority of survey respondents (85.2%) live in the urban space…”

L181 – Change to “(14.8%) answers, with the following…” since with is misspelled

L185-191 – To be more concise and clear, change the last four sentences in this paragraph to “Most survey respondents were married with a Master’s Degree and/or Bachelor’s Degree living with two to four people in their household (Table 2).” Make sure to delete the last sentence as this is now redundant.

L193-194 – Change to “…key factors for consumer buying behavior.”

L194 – Change to “As shown in Figure 3, we generated a biplot…”

L198 – Add another sentence to this paragraph at the end to better clarify what is being graphed…“In these type of graphs, the four quadrants represent different combinations of sociocultural characteristics we surveyed.” As stated in the initial review, this is unclear and needs to be clarified for the reader.

L199-200 – Note that in Figure 3 the word “Married” is misspelled

L210 – To be consistent elsewhere with the edits, change to “households of four or five people, and”

L215 – Change to “living with four to six people in the same household.”

L216 – Change to “from families of two or three people”

L221 – How is after the crisis and post crisis different? Change to “…behavior before, during, and post crisis.”

L220-230 – These are two paragraphs…merge into ONE paragraph and if you are putting Table 3 between the paragraph then do NOT indent the second part.

L229 – Remove comma so change to “…producers within the…”

L247 – Change to “13 (5.05%) people bought…”

L258 – Change to “of vegetable consumption,”

L258-272 – These are two paragraphs…merge into ONE paragraph.

L270-272 – Change so more direct statement so change to “Interestingly, only one respondent of the 31 total respondents expressed no interest in…” 

L275 – Change to “…towards alternative ways of purchasing produce.”

L289 – Change to “The emotional state of people living in quarantined areas [36-38] leads us…”

L291 – Change to “As shown in Figure 8, the”

L301 – Change to “…can provide additional insight to further analyze this”

L309-310 – Change to “…buyers will generally adhere to this pattern…”

L310-313 – Make sentence more direct by changing to “Weekly purchase of produce is the most dynamic and a key vector for enhancing…”

L315 – Start new paragraph here with sentence “Another interesting aspect…” the start of this new paragraph

L320 – Start new paragraph here with sentence “Given the significant number of…” the start of this new paragraph

L323 – Change to “by weekly home delivery. Cash will remain the favorite payment method. If”

L323-327 – Simplify this last sentence so change to “Survey respondents who purchased or plan on purchasing locally before, during, and after the COVID-19 crisis have similar percentage values for frequency of these purchases (Figure 10).”

L333-337 – Simplify this first sentence so change to “For survey respondents purchasing locally before (31 replies), during (154 replies), and who plan on purchasing after (207 replies) the crisis, preference for weekly orders dropped while preference for ordering every two weeks increased.”

L339-342 – Consolidate these two sentences into one and keep to this being a suggested inference as the survey did not ask directly about food waste. Change to “This behavior shift of purchasing only when necessary suggests consumers may be more sensitive to food waste and are aware of the high perishability of fresh vegetables.” Make sure to delete the sentence that states consumers became more aware of food waste.

L343-345 – Consolidate these two sentences so change to “Accordingly, consumer interest increased for local, fresh, and highly nutritious food which could impact quality of life for consumers.”

L349 – Change to “SFSC”

L354-373 – These are three paragraphs…merge into ONE paragraph and if you are putting Table 4 between the paragraph then do NOT indent the second and third parts.

L354 – Consolidate the first two sentences so change to “Table 4 presents frequencies of online or face-to-face channels preferred by consumers purchasing local produce.”

L367 – Change to “These three categories…”

L372-373 – Change to “…and ignored diversifying online marketing platforms.”

L382-390 – These are three paragraphs…merge into ONE paragraph and if you are putting Table 5 between the paragraph then do NOT indent the second and third parts.

L391-400 – These are two paragraphs…merge into ONE paragraph. The topic header sentence does not need to stand alone and can be the first sentence in the next paragraph. Basically a paragraph by definition is three sentences or more. This forces the writer to better organize the structure of the writing with a topic first sentence followed by supporting arguments supporting the topic sentence.

L395 – To be consistent, insert a space between so “total of 257 (14.4%);”

L395-396 – Change to “This may be explained…”

L398 – Change to “…is also supported by Romanian consumers symbolically associating agrifood products with childhood memories or…”

L404 – Remove one comma so change to “chains, in particular fresh vegetables from producer, there...”

L405 – Change to “and platforms in Romania. Among existing online platforms, there are even fewer e-commerce apps”

L406-407 – Change last sentence to “This suggests a high degree of wishful thinking by survey respondents”

L411-420 – These are three paragraphs…merge into ONE paragraph and if you are putting Table 6 between the paragraph then do NOT indent the second and third parts.

L411 – Misspelled are so change to “…options are included…”

L413 – Change to “Table 6 presents the frequencies…”

L417 – Make the writing more straight to the point without repeating the analysis topic so change to “Even under the restrictive circumstances”

L432-450 – These are a hanging solo sentence and a paragraph…merge into ONE paragraph and put Table 7 and Figure 13 after this merged paragraph.

L439 – Change to “As shown in Table 7, most respondents (121 respondents, 47.08%) have chosen cash payment,”

L440 – Change to “debit card” as there is no reason to capitalize this

L440-441– Change to “(113 respondents, 43.97%), while bank transfer is the least preferred payment method (23 respondents, 8.95%).”

L443 – Change to “…use the cash payment…”

L446 – Change to “against the COVID-19 epidemic.”

L449 – Change to “delivery, local producers…”

L450 – Change to “card using Square card app and swiper or facilitating electronic bank transfers.”

L451-473 – These are a hanging solo sentence and a paragraph…merge into ONE paragraph and put Figure 14 after this merged paragraph or in between but make sure not to indent the second section.

L451 – Change to “We again use multivariate correspondence to analyze the”

L453 – Change to “…by the new coronavirus (Figure 14).”

L463 – Change to “…prefer weekly purchases. Since ‘After the COVID-19 crisis – Yes’ is located”

L467 – Start new paragraph with sentence “Accordingly, there are…”

L469 – Change to “remains at the top of preferred payment methods.”

L471-473 – Change last sentence to “The ideal customer purchasing local produce are established customers aged 35-49 buying one or twice a week who prefer paying cash and will keep buying post-crisis.”

L474-490 – These are a hanging solo sentence and two paragraphs…merge into ONE paragraph.

L474 – Add a topic sentence to this newly merged paragraph so add “Other quadrants in the graph correspond to other customer typologies. Quadrant 2 (Q2) corresponds to those who order”

L475 – Change to “…from producers monthly, who have not ordered”

L477 – Change to “Quadrant 3 is associated with people who did not order”

L480-481 – Change to “It is highly likely that this profile corresponds to retired people…”

L482 – Change to “These elderly people…” and change to “…vegetable at farmers’ markets located near”

L487 – Change to “Quadrant 4 is associated with consumers who ordered fresh”

L489 – Change to categories from Q1, namely preference for”

L491-505 – These are a hanging solo sentence and a paragraph…merge into ONE paragraph.

L491 – Make this the topic sentence to the next paragraph and change to “Several aspects of age cluster data distribution warrant further consideration.”

L496-499 – Merge these two sentences so more direct and clear so change to “This suggests the inertia of traditional buying behavior in short food supply chains (SFSCs) is rather higher for older consumers.”

L500-501 – Change to “…from producers on SFSCs.”

L501-502 – Change to “As we have previously mentioned, such elderly motivations could reside in culture or could be related to socializing needs of this age group.”

L503-505 – Make sure NOT to capitalize cash and debit card so written as “cash” and “debit card”

L506-508 – Change to “The density of values are represented most closely to the junction of both axes in our multivariate correspondence graphs. Cash or debit card payments show high densities which are close in terms of percentage.”

L511-512 – Change to “…short food supply chains that are currently in an emergent phase in Romania.”

L514-521 – These are a hanging solo sentence and a paragraph…merge into ONE paragraph.

L517-518 – Change to “However, consumer avoid shopping at grocery stores, farmers’ markets and/or supermarkets which were often crowded during the pandemic.”

L518-519 – Change to “Social distancing is respected and contact with unknown, possibly infected people is avoided.”

L519-521 – This sentence does not make any sense…How can asymptomatic consumers have no contact with markets? This was not ascertained in your survey. Please delete this sentence or make clearer.

L522-524 – Change this sentence to “To increase local food production and sales in Romania, small local producers need to adhere to shifting customer preferences and innovate their marketing strategies.”

L529 – Change to “times of crises and beyond.”

L530-549 – Write the last two paragraphs not in numerical order of recommendations but in more of a written format making connections between the points.

L551-553 – Change first sentence to “Our results confirms the hypothesis that the COVID-19 pandemic induced significant changes in consumer purchasing behavior of fresh vegetables.”

L552 – Change to “…consumers are more determined to”

L557 – Change to “this system of buying from short food supply chains (SFSCs) after”

L561 – Change to “…transformation of SFSCs.”

L566-575 – These are two short paragraphs…merge into ONE paragraph.

L568-569 – Change to “Thus SFSCs represent…”

L574 – Change to “SFSCs” to be consistent

L576-577 – Change to “…is tied to the current preliminary study of just the quarantined area of Suceava.”

L579-580 – Change to “…the results of our study will add to the growing body of research on short food supply chains conducted nationwide in Romania as well as globally.”

Author Response

This manuscript analyzes the impacts of the COVID-19 pandemic on consumer purchasing patterns for vegetables from local farmers in a county in Romania. However, there are substantive short-comings of the manuscript that as a reviewer I cannot recommend publication at this time unless FOUR substantive changes described below and line number specific edits are made. The first substantive change was not addressed from the initial review. This reviewer would be open to re-evaluating the manuscript after these changes are made:

Our response: Thank you for your revision and for your availability too. In this new revised manuscript we have tried to address all your suggestions.

1) The first part of the discussion should contrast the results from the study with prior literature on impacts on food supply chains from historical disruptions similar to COVID-19 (e.g. 1918-19 Spanish flu, etc.). This will also increase the number of citations as the number currently is low. This was requested during the first review but not done. The research needs to show how it fits into the broader literature. Since COVID-19 is currently happening, the contrasts of the research results with other literature are likely to be literature on similar disruptions to food marketing from other historical events in Romania and elsewhere in the world.

Our response: In the first part of Chapter 4 entitled Discussions we have inserted the following paragraphs:

There are numerous authors who have already insisted on the consequences brought by pandemics on the economy and, especially, on the product distribution systems [70-75]. Some of these support the idea that such sanitary crises were followed by economic growth as a direct consequence of the consumption increase [76,77], while others say that on the contrary, the effects are negative for the human activities [78-80] and, especially for agriculture [81]. Coming back to the nowadays pandemic, according to Carlsson-Szlezak [82-83], there are three types of effects COVID-19 has, namely an impact upon consumption, one upon the market, and another one on the distribution chains. This is the main reason why we think that the food distribution systems should be reconsidered to strengthen the resilience and, in the future, address all complex sides of the contemporary society [84-87].

2) The last part of the discussion needs to add a paragraph at the end on how to address the limits agricultural producers and consumers face when it comes to following these recommendations that have been made in the prior two paragraphs. That is really at the heart of the discussion so if this is more than one paragraph that is no problem. The end of the currently written discussion offers recommendations based on the results. That is great but the discussion needs to go beyond this and discuss the implications on how practical these recommendations are and how any barriers and challenges to following these recommendations can be substantively addressed.

Our response: In the first part of Chapter 4 entitled Discussions we have inserted the following paragraphs:

For business development, certain limitations need to be taken into consideration. Firstly, supply with fresh vegetables directly from local producers cannot be achieved in Romania throughout the calendar year due to the seasonal nature of crops and reduced areas designated for greenhouses or poly tunnels [88,89]. The quantities and vegetable varieties locally produced and available on the market are reduced due to their zonal and seasonal nature, and, consequently, the demand cannot be exclusively covered by the local production [88]. Another key factor is the final cost of the product, as the price of the local vegetables sold through SFSCs is regularly higher than the prices of similar products in the hypermarket networks [90,91]. To address these issues and come up with viable solutions, the producers could follow the successful associative models from the Western Europe thus gaining more visibility and authority on the market. However, the reticence of Romanian small producers is a common denominator when it comes to associating and cooperating locally, including making a common brand, which cannot but become a  hindrance for penetrating the local market [92–94]. 

             In Romania, the digital transformation of small producers can have a positive effect for the entire economy as the implementation of online commerce technologies and adjusting the payment methods for debit card și bank transfer can lead to increasing the taxation degree of the sales. However, the digital transformation is also influenced by certain local factors. For instance, to develop distribution channels, small producers need to invest in infrastructure. Nevertheless, their financial possibilities are rather limited and, they choose to invest in means of production at the expense of infrastructure for commercialization and marketing [95]. Furthermore, many small enterprises are in dire need of time, another impediment for digital transformation as they allot most of their activities to production and placing it on the market. Additionally, the scarce digital literacy of the producers is a barrier preventing and limiting the development of the innovative instruments of SFSC [96].

3) After conducting more literature search on this, there may be enough information to add a paragraph to the introduction on historical impacts to agricultural marketing during prior pandemics or disruptions to introduce the reader to the context of the analysis.

Our response: We have reconsidered the text of Chapter 1 entitled Introduction and we have added the paragraphs below:

Pandemics are not exactly a novel phenomenon strictly related to the nowadays societies as they were recorded since ancient times. Each pandemic triggered major changes in economics, regional and global policies, social behavior, and mentalities as well. The most significant changes (which have been preserved on medium and long term) have been those institutionalized [1]. By contrast, the changes which were least preserved are related to mentalities and social behavior as the institutionalized modifications [2], through public policies, were not sufficiently coupled and consolidated with the psychosocial changes [3]. As any other pandemics, COVID-19 has caused significant changes on all levels of contemporary society [4–9]. All states, continents, regions, urban and rural communities, families, and, ultimately, thinking and lifestyle of each individual have been impacted by the pandemic [10–14]. And we may never return to the normality previously experienced, before COVID-19 [15,16].

At the same time, each pandemic in the recorded history had immediate effects on the primary reactions of the social human, because they affected directly the concern for health, financial security, life quality and food security [17]. For instance, when cholera or the Spanish flu hit, the economic balance and food supply systems broke and caused famine as an immediate effect [18]. The COVID-19 pandemic has largely fit the same profile, although there are specific differences. This time there have not been recorded major negative effects on the food security, except for the underdeveloped and developing countries, while the developed economies have not faced serious problems in terms of global food security [19,20]. Inherently, there were individual problems, especially in the case of quarantined people with low and very low income [21]. Nevertheless, the concern for food security has turned into concern for food safety as the public space of the developed countries was invaded by issues such as healthy eating. 

However, all pandemics share the demographic vector of disease spread. During Middle Ages, the pandemic would be transmitted from one part of the Europe to the other through the people who fled the outbreak [18]. The Amerindian population were decimated by the diseases brought by the European explorers, as they lacked any inherited immunity to the infectious diseases of Europe (in this case, Tzvetan Todorov says that the first globalization was the spread of the viruses [22]). The Spanish flu spread mainly due to the movement of the soldiers from the WW II (as they came home back in 1918, they spread the pandemic globally) [23]. The COVID-19 pandemic has been largely triggered by population density [24,25], high degree of mobility of the contemporary humans and mass socialization, by cultural, social, and tourism events [26–28]. Consequently, the measures taken by most world states have addressed issues such as quarantine and isolation, more precisely the enforced social isolation of the population along with the economic isolation between various states or regions as well as between different economic sectors [29]. Hence, this lockdown has impeded the interactions among food systems incorporating every stage of food production and delivery.

If we are to look into Romania’s case, several differences between COVID-19 and the other pandemics should be underlined. For instance, the Spanish flu, by its direct and indirect effects (which are quite difficult to assess), overlapped the lack of organizational deficiencies of the primary sector, at that time the fundamental branch for achieving the national income (a statistical indicator of the period which is equal to GDP). The census of 1930 Romania estimated that approximately 50% of the material production was provided by the rural population. Therefore, under the circumstances where nearly three quarters of the population was working in agriculture, it is understandably that the Spanish flu had a catastrophic effect on the civil population and army, and contributed to the disruption of economic activities. Additionally, in that period, most Romanian population had a food diet which could be regarded as unbalanced by comparison with the inhabitants of the Western states of Europe. More precisely, monophagism or excessive consumption of cereals had largely contributed to the aggravation of effects of the Spanish flu due to, among others, a weakened immune system. However, in Romania, the COVID-19 pandemic has occurred in a totally different socioeconomic context from the Spanish flu [30–32].  

4) What is the difference between 12.06% and 12%? Rounding off does not change the point that is being made. Including that many decimal places actually increases the difficulty of interpreting results. Please round off to the nearest percent throughout the written manuscript.

Our response: We have taken into account your suggestion and we made the necessary corrections. More precisely, we revised all the numerical values in the shape of “X .51%” by rounding them up, the end result being a numerical value of “X+1%”. All numbers “Y” with the first decimal place between 1-5 were rounded down, the end result being “Y”. All the necessary changes were made in the text of the manuscript wherever needed, especially where the analysis of the results of our questionnaire was referred to. Furthermore, Figure 4 was altered according to the aforementioned suggestions.

Specific Line Number of Manuscript edits (note that requested change of word(s) in quotations) and places denoted by “???” where the co-authors will have to write in to clarify:

L30 – Remove “the”

Our response: We have made the suggested modification

L51 – Remove “it is”

Our response: We have made the suggested modification.

L56-62 – These are two paragraphs…merge into ONE paragraph

Our response: We have made the suggested modification

L66 – Change to “The population distribution of age groups,”

Our response: We have replaced “on” with “of”

L71 – Change to “Based on the sociocultural characteristics mentioned above,”

Our response: We have replaced “Based on the features mentioned above“ with “Based on the sociocultural characteristics mentioned above“

L73 – Change to “short food supply chains (SFSC).”

Our response: We have inserted the acronym SFSC in the text.

L75 – Change to “during the COVID-19 crisis,”

Our response: We have replaced “during COVID -19 crisis” with “during the COVID-19 crisis”

L80-86 – These are two paragraphs…merge into ONE paragraph

Our response: We have made the suggested modification

L87-88 – Change to “To test this hypothesis, our research identified sociocultural factors impacting consumer purchasing decisions before the COVID-19 crisis.”

Our response: We have changed “To test this hypothesis, our research has started from identifying the sociocultural factors with impact upon the purchasing decision before COVID-19 crisis “ to ”To test this hypothesis, our research identified sociocultural factors impacting consumer purchasing decisions before the COVID-19 crisis“.

L89 – Change to “how the quarantine influenced consumer buying behavior…”

Our response: We have changed “how the quarantine influenced the consumer buying behavior “ to  “how the quarantine influenced consumer buying behavior”.

L90 – Change to “The narrative-argumentative sketch of this analysis is outlined in Figure 2.”

Our response: We have changed “is described” to “is outlined”

L91-92 – Reduce the size of Figure 2 so it fits in the blank part of the previous page (you will have to do this of page proof edits)

Our response: Despite having made the necessary alterations for reducing the size of Figure 2, as suggested, still we are not pretty sure how it will fit on the page in the final version of the manuscript.

L94-100 – To be consistent with other parts of the manuscript, please use numbers and write out number of questions…for example, change to “as follows: 1) eight questions…” and “location), 2) three questions…” and “the crisis, 3) one question…” and “by respondents, and 4) five questions…”

Our response: We have changed the figures with words : 8 to eight, 3 to three, and so on.

L104 – Change to “…were made available...”

Our response: We have changed “was made available” to “were made available”

L106-115 – These are three paragraphs…merge into ONE paragraph

Our response: We have made the suggested modification.

L115 – Change to “of short food supply chains as well” and

Our response: We have made the suggested modification, ie changed “of the SFSC as well” to “of short food supply chains as well”

L117 – Change to “…related to short food supply chains (SFSCs), local producers, food”

Our response: We have  changed “..related to short food supply chains, local producers, food” to “..  related to short food supply chains (SFSCs), local producers, food”

L120 – Change to “…upon the SFSCs.”

Our response: We have changed “…upon the short food supply chains” to “upon the SFSCs “

L127 – Change to “…namely SFSCs, food hubs, local agrifood”

Our response: We have changed “namely SFSC” to “namely SFSCs”

L129 – Change to “Short food chain supply (SFSC) systems…”

Our response: We have changed “SFSC system“ to “Short food chain supply (SFSC) systems”.

L130 – Change to “health nature) for people and society as a whole: new job…”

Our response: We have changed “natural persons” to “people“.

L133 – Change to “…preserving cultural heritage, including…”

Our response: We have removed the word “the”, as suggested.

L134 – Change to “…securing access to healthier”

Our response: We have changed “securing the access “ to “securing access”

L136 – Change to “At the same time, SFSC systems…”

Our response: We have made the suggested modification.

L146-147 – Change to “…between potential buyers and local producer(s) in a certain region.”

Our response: We have changed “between potential buyer and local producer ´to “…between potential buyers and local producer(s)”.

L153-158 – Change to “Changing consumption patterns, holding…” Also, break up this very long sentence into two separate sentences to improve the reader’s ability to process this so change “…fresh food, ellipses in the agro-industrial sector, and exponential increase of online deliveries are impacts from the pandemic in agriculture for most European states, Romania included. Potential challenges include imposing restrictions on the movement of goods, lack of labor force due to the enforced quarantine, closing down important economic agents such as HORECA (Hotels / Restaurants / Café) network, as well as the temporary closure of schools and cafeterias [33].”

Our response: We have made the suggested alterations. The paragraph looks like this now:

“Changing consumption patterns, holding excess inventory in the commercial chains and by customers as well changing the percentage of basic food products/ fresh food, ellipses in the agro-industrial sector, and exponential increase of online deliveries are impacts from the pandemic in agriculture for most European states, Romania included. Potential challenges include imposing restrictions on the movement of goods, lack of labor force due to the enforced quarantine, closing down important economic agents such as HORECA (Hotels / Restaurants / Café) network, as well as the temporary closure of schools and cafeterias”

L157 – You need to state what this HORECA stands for so change to “such as ??? (HORECA) network, as well…”

Our response: We have made the suggested alterations by substituting the term with “Hotels/Restaurants/Café”.

L175-191 – These are three paragraphs…merge into ONE paragraph and if you are putting Table 2 between the paragraph then do NOT indent the second part.

Our response: We`ve acted accordingly and taken into account your suggestion by merging the aforementioned paragraphs. Moreover, Table no. 2 was placed in between the respective paragraphs. As for the second part of the paragraph we removed the indentation mentioned.

L179 – Also, please do not start a sentence with a numerical percentage so change to “The majority of survey respondents (85.2%) live in the urban space…”

Our response: As suggested, we made the necessary alteration and rounded down the figure 85.2% to 85%.

L181 – Change to “(14.8%) answers, with the following…” since with is misspelled

Our response: We have made the suggested alteration by rounding up the number 14.8% to 15%, and corrected the word “with”

L185-191 – To be more concise and clear, change the last four sentences in this paragraph to “Most survey respondents were married with a Master’s Degree and/or Bachelor’s Degree living with two to four people in their household (Table 2).” Make sure to delete the last sentence as this is now redundant.

Our response: We have made the suggested modification.

The paragraph

“85.2% of the respondents of the survey live in the urban space of the quarantined area, Suceava — 215 (83.7%) and Salcea — 4 (1.6%). From the rural space of the quarantined area we received 38 (14.8%) answers, whith the following distribution by locality: Adâncata (3 – 1.2%), Bosanci (1 – 0.4%), Ipotești (14 – 5.4%), Moara (3 – 1.2%), Șcheia (13 – 5.1%), Pătrăuți (1 – 0.4%), and Mitocu Dragomirnei (3 – 1.2%).

According to gender, out of the total respondents (N=257), most answers (87%) come from women. In the same time, most answers are in the age categories of 20-34 and 35-49. About the marital status, most answers have come from married persons. Related to the educational background, most answers are in the following categories: Bachelor's Degree and Master’s Degree. On the question “How many people live in your household”, most answers indicated 2 persons, 3 persons, and 4 persons (Table 2). Based on these frequencies, the general profile of the respondent to this survey can be outlined as follows: married women of 20-49 years old who live in households of 3 or 4 persons, and have a Bachelor’s or Master’s Degree.”

Is now changed to

“The majority of survey respondents (85%) live in the urban space of the quarantined area, Suceava — 215 (84%) and Salcea — 4 (2%). From the rural space of the quarantined area we received 38 (15%) answers, with the following distribution by locality: Adâncata (3 – 1.2%), Bosanci (1 – 0.4%), Ipotești (14 – 5.4%), Moara (3 – 1.2%), Șcheia (13 – 5.1%), Pătrăuți (1 – 0.4%), and Mitocu Dragomirnei (3 – 1.2%). According to gender, out of the total respondents (N=257), most answers (87%) come from women. In the same time, most answers are in the age categories of 20-34 and 35-49. Most survey respondents were married with a Master’s Degree and/or Bachelor’s Degree living with two to four people in their household (Table 2).”

L193-194 – Change to “…key factors for consumer buying behavior.”

Our response: We have changed  “key factors for the consumer buying behavior” to “key factors for consumer buying behavior”.

L194 – Change to “As shown in Figure 3, we generated a biplot…”

Our response: We have changed “In Figure 3 it is used a biplot “ to  “As shown in Figure 3, we generated a biplot… “ .

L198 – Add another sentence to this paragraph at the end to better clarify what is being graphed…“In these type of graphs, the four quadrants represent different combinations of sociocultural characteristics we surveyed.” As stated in the initial review, this is unclear and needs to be clarified for the reader.

Our response: We have made the suggested modification.

L199-200 – Note that in Figure 3 the word “Married” is misspelled

Our response: We have corrected the word “married”.

L210 – To be consistent elsewhere with the edits, change to “households of four or five people, and”

Our response: We have made the suggested modification, and replaced figures with words 4 to “four”, 5 to “five”.

L215 – Change to “living with four to six people in the same household.”

Our response: We have made the suggested modification, and replaced figures with words

L216 – Change to “from families of two or three people”

Our response: We have made the suggested modification.

L221 – How is after the crisis and post crisis different? Change to “…behavior before, during, and post crisis.”

Our response: We have made the suggested modification and changed “after “with “during”.

L220-230 – These are two paragraphs…merge into ONE paragraph and if you are putting Table 3 between the paragraph then do NOT indent the second part.

Our response: We`ve acted accordingly and taken into account your suggestion by merging the aforementioned paragraphs. Moreover, Table no. 3 was placed in between the respective paragraphs. As for the second part of the paragraph we removed the indentation mentioned.

L229 – Remove comma so change to “…producers within the…”

Our response: We have made the suggested modification.

L247 – Change to “13 (5.05%) people bought…”

Our response: As suggested we made the necessary change. The second decimal place was rounded down (5%).

L258 – Change to “of vegetable consumption,”

Our response: We have changed “of the vegetable consumption “ to  “of vegetable consumption“.

L258-272 – These are two paragraphs…merge into ONE paragraph.

Our response: We have made the suggested modification.

L270-272 – Change so more direct statement so change to “Interestingly, only one respondent of the 31 total respondents expressed no interest in…”

Our response: We have made the suggested modification, thus changed “An interesting fact is that, out of the total respondents, merely one respondent has shown no interest in” to “Interestingly, only one respondent of the 31 total respondents expressed no interest in…”.

L275 – Change to “…towards alternative ways of purchasing produce.”

Our response: We have changed “..supply..” with “..purchasing  produce…”.

L289 – Change to “The emotional state of people living in quarantined areas [36-38] leads us…”

Our response: We have changed “… The emotional state of persons quarantined areas ..” to “The emotional state of people living in quarantined areas…”.

L291 – Change to “As shown in Figure 8, the”

Our response: We have made the suggested modification, and changed “As it is visible in Figure 8, the” to “As shown in Figure 8, the”.

L301 – Change to “…can provide additional insight to further analyze this”

Our response: We have made the suggested modification, and changed “...can provide extra interpretation data to further analyze this..” to “….can provide additional insight to further analyze this..”

L309-310 – Change to “…buyers will generally adhere to this pattern…”

Our response: We have made the suggested modification, and changed “..buyers will generally follow up this pattern..” to “…buyers will generally adhere to this pattern…” .

L310-313 – Make sentence more direct by changing to “Weekly purchase of produce is the most dynamic and a key vector for enhancing…”

Our response: We have made the suggested modification.

L315 – Start new paragraph here with sentence “Another interesting aspect…” the start of this new paragraph

Our response: We have made the suggested modification and split the paragraph.

L320 – Start new paragraph here with sentence “Given the significant number of…” the start of this new paragraph

Our response: We have made the suggested modification.

L323 – Change to “by weekly home delivery. Cash will remain the favorite payment method. If”

Our response: We have made the suggested modification and shortened the phrase as follows “by weekly home delivery, while  Cash will remain the favorite payment” was changed to “…by weekly home delivery. Cash will remain the…..”.

L323-327 – Simplify this last sentence so change to “Survey respondents who purchased or plan on purchasing locally before, during, and after the COVID-19 crisis have similar percentage values for frequency of these purchases (Figure 10).”

Our response: We have changed “If we are to separate into clusters and distribute the affirmative answers on three categories (those who bought before March 16, those who bought after March 16, and those who state that they will buy after the end of COVID-19 crisis), we shall notice close percentage values in the preferences shown for the purchase frequencies of fresh vegetables directly from producers (Figure 10).“

 to

“Survey respondents who purchased or plan on purchasing locally before, during, and after the COVID-19 crisis have similar percentage values for frequency of these purchases (Figure 10).”

L333-337 – Simplify this first sentence so change to “For survey respondents purchasing locally before (31 replies), during (154 replies), and who plan on purchasing after (207 replies) the crisis, preference for weekly orders dropped while preference for ordering every two weeks increased.”

Our response: We`ve taken action to simplify the first sentence, according to your suggestion.

L339-342 – Consolidate these two sentences into one and keep to this being a suggested inference as the survey did not ask directly about food waste. Change to “This behavior shift of purchasing only when necessary suggests consumers may be more sensitive to food waste and are aware of the high perishability of fresh vegetables.” Make sure to delete the sentence that states consumers became more aware of food waste.

Our response: We have consolidated the aforementioned sentences. Furthermore, we made sure that the indicated sentence was eliminated.

L343-345 – Consolidate these two sentences so change to “Accordingly, consumer interest increased for local, fresh, and highly nutritious food which could impact quality of life for consumers.”

Our response: We have made the suggested modification.

L349 – Change to “SFSC”

Our response: We have corrected “SFCS” to “SFSC”.

L354-373 – These are three paragraphs…merge into ONE paragraph and if you are putting Table 4 between the paragraph then do NOT indent the second and third parts.

Our response: We merged all three paragraphs labeled L354-374 and inserted them above Table 4.

L354 – Consolidate the first two sentences so change to “Table 4 presents frequencies of online or face-to-face channels preferred by consumers purchasing local produce.”

Our response: We have deleted the first two sentences and made the suggested modification.

L367 – Change to “These three categories…”

Our response: We have replaced 3 to “three”.

L372-373 – Change to “…and ignored diversifying online marketing platforms.”

Our response: We have changed “and ignored the resilience strategies for possible risk factors“ to “and ignored diversifying online marketing platforms”.

L382-390 – These are three paragraphs…merge into ONE paragraph and if you are putting Table 5 between the paragraph then do NOT indent the second and third parts.

Our response: As indicated we have merged the respective paragraphs ranging L382-390. Then, we placed before Table 5.

L391-400 – These are two paragraphs…merge into ONE paragraph. The topic header sentence does not need to stand alone and can be the first sentence in the next paragraph. Basically a paragraph by definition is three sentences or more. This forces the writer to better organize the structure of the writing with a topic first sentence followed by supporting arguments supporting the topic sentence.

Our response: As indicated we took action in this matter. As a result, we have merged the paragraphs ranging L391 to L400.

L395 – To be consistent, insert a space between so “total of 257 (14.4%);”

Our response: We have made the suggested modification.

L395-396 – Change to “This may be explained…”

Our response: We have changed “this fact may be explained” to “This may be explained…”.

L398 – Change to “…is also supported by Romanian consumers symbolically associating agrifood products with childhood memories or…”

Our response: We have changed “is also supported by the fact that, in the case of many Romanian consumers the agrifood products are symbolically associated with childhood memories” to “is also supported by Romanian consumers symbolically associating agrifood products with childhood memories or…”.

L404 – Remove one comma so change to “chains, in particular fresh vegetables from producer, there...”

Our response: We have made the suggested modification.

L405 – Change to “and platforms in Romania. Among existing online platforms, there are even fewer e-commerce apps”

Our response: We have shortened the sentence by removing “and” and starting a new sentence, as suggested above.

L406-407 – Change last sentence to “This suggests a high degree of wishful thinking by survey respondents”

Our response: We have made the suggested modification.

L411-420 – These are three paragraphs…merge into ONE paragraph and if you are putting Table 6 between the paragraph then do NOT indent the second and third parts.

Our response: As suggested we merged the paragraphs ranging L413 to L420 and reinserted them before Table 6. Furthermore, we moved the paragraph ranging L411-412 before Figure 12 given the fact that upon our latest review of the manuscript we observed that the content of the respective paragraphs is linked to the previous ones.

L411 – Misspelled are so change to “…options are included…”

Our response: We have made the suggested modification, and changed “ere” to “are”

L413 – Change to “Table 6 presents the frequencies…”

Our response: We have made the suggested modification, and changed “In Table 6 there are presented the frequencies” to “Table 6 presents the frequencies…”

L417 – Make the writing more straight to the point without repeating the analysis topic so change to “Even under the restrictive circumstances”

Our response:  As suggested we cropped the “Even under the restrictive circumstances” part and then we inserted it at the beginning of the aforementioned section. Upon our latest revision we aimed at solving this issue by eliminating any redundancies.

L432-450 – These are a hanging solo sentence and a paragraph…merge into ONE paragraph and put Table 7 and Figure 13 after this merged paragraph.

Our response: As indicated, we acted accordingly by merging the respective paragraphs into one. Then we made the necessary alterations regarding both Table 7 and Figure 13.

L439 – Change to “As shown in Table 7, most respondents (121 respondents, 47.08%) have chosen cash payment,”

Our response: We have made the suggested modification.

L440 – Change to “debit card” as there is no reason to capitalize this

Our response: We have made the suggested modification.

L440-441– Change to “(113 respondents, 43.97%), while bank transfer is the least preferred payment method (23 respondents, 8.95%).”

Our response: We have made the suggested modification.

L443 – Change to “…use the cash payment…”

Our response: We have made the suggested modification.

L446 – Change to “against the COVID-19 epidemic.”

Our response: We have made the suggested modification.

L449 – Change to “delivery, local producers…”

Our response: We have made the suggested modification.

L450 – Change to “card using Square card app and swiper or facilitating electronic bank transfers.”

Our response: Aiming at being more precise, we followed your suggestion and we made the necessary changes: “Square Point of Sale (POS) or facilitating electronic bank transfers.”

L 439-L450 Most part of the respondents (47.08%) have chosen the Cash payment (121 respondents), followed by Debit Card payment (113 respondents, 43.97%). The bank transfer is the less preferred option as payment method (23 respondents, 8.95%) (Table 7). If we analyze only the answers received from persons choosing home delivery (before, during and post crisis) it can be noticed that the percentage of respondents who use the Cash payment has been significantly reduced in favor of card payments or bank transfers (Figure 13). This demonstrates that during COVID-19 crisis the population became aware that card payments or bank transfers can be safe preventive measures against COVID-19 epidemic. At the same time, this type of behavior is supported by the fact that over this period of crisis, buyers lack cash, since the regulations enforced by the state of emergency have greatly decreased ATM cash withdrawals. To be able to commercialize fresh vegetables with home delivery, the local producers need to adjust and facilitate their customers’ payments made by debit card or bank transfer.

Changed to

As shown in Table 7, most respondents (121 respondents, 47%) have chosen cash payment, (113 respondents, 44%), while bank transfer is the least preferred payment method (23 respondents, 9%). If we analyze only the answers received from persons choosing home delivery (before, during and post crisis) it can be noticed that the percentage of respondents who use the cash payment has been significantly reduced in favor of card payments or bank transfers (Figure 13). This demonstrates that during COVID-19 crisis the population became aware that card payments or bank transfers can be safe preventive measures against the COVID-19 epidemic. At the same time, this type of behavior is supported by the fact that over this period of crisis, buyers lack cash, since the regulations enforced by the state of emergency have greatly decreased ATM cash withdrawals. To be able to commercialize fresh vegetables with home delivery, local producers need to adjust and facilitate their customers’ payments using Square Point of Sale (POS) or facilitating electronic bank transfers.

L451-473 – These are a hanging solo sentence and a paragraph…merge into ONE paragraph and put Figure 14 after this merged paragraph or in between but make sure not to indent the second section.

Our response: As indicated we revised and reconsidered the text. More precisely, we left Figure 14 below the paragraph ranging lines 451 to 453, given the fact it is an introductory paragraph referring to the aforementioned figure. Moreover, we then merged the paragraphs ranging lines 459-474 located below Figure 14.

L451 – Change to “We again use multivariate correspondence to analyze the”

Our response: We have made the suggested modification.

L453 – Change to “…by the new coronavirus (Figure 14).”

Our response: We have made the suggested modification.

L463 – Change to “…prefer weekly purchases. Since ‘After the COVID-19 crisis – Yes’ is located”

Our response: We have made the suggested modification.

L467 – Start new paragraph with sentence “Accordingly, there are…”

Our response: We have made the suggested modification.

L469 – Change to “remains at the top of preferred payment methods.”

Our response: We have made the suggested modification.

L471-473 – Change last sentence to “The ideal customer purchasing local produce are established customers aged 35-49 buying one or twice a week who prefer paying cash and will keep buying post-crisis.”

Our response: We have made the suggested modification.

L474-490 – These are a hanging solo sentence and two paragraphs…merge into ONE paragraph.

Our response: We have made the suggested modification.

L474 – Add a topic sentence to this newly merged paragraph so add “Other quadrants in the graph correspond to other customer typologies. Quadrant 2 (Q2) corresponds to those who order”

Our response: We have made the suggested modification.

L475 – Change to “…from producers monthly, who have not ordered”

Our response: We have made the suggested modification and changed “…from producers on a monthly basis (once a month), who have not ordered..” to “…from producers monthly, who have not ordered”

L477 – Change to “Quadrant 3 is associated with people who did not order”

Our response:  “In the quadrant 3 (Q3) there are the most important correlations among those who did not order..”  changed as suggested.

L480-481 – Change to “It is highly likely that this profile corresponds to retired people…”

Our response:  “It is highly likely that this profile regards mostly retired people “ was changed as suggested above.

L482 – Change to “These elderly people…” and change to “…vegetable at farmers’ markets located near”

Our response: We have made the suggested modification and changed “These elderly persons normally go shopping for fresh vegetables to farmers’ markets located near their homes” as suggested above.

L487 – Change to “Quadrant 4 is associated with consumers who ordered fresh”

Our response: We have changed the first sentence of the paragraph as suggested.

L489 – Change to categories from Q1, namely preference for”

Our response: We have made the suggested modification.

L491-505 – These are a hanging solo sentence and a paragraph…merge into ONE paragraph.

Our response: We have made the suggested modification.

L491 – Make this the topic sentence to the next paragraph and change to “Several aspects of age cluster data distribution warrant further consideration.”

Our response: We have made the suggested modification.

L496-499 – Merge these two sentences so more direct and clear so change to “This suggests the inertia of traditional buying behavior in short food supply chains (SFSCs) is rather higher for older consumers.”

Our response: We have made the suggested modification.

L500-501 – Change to “…from producers on SFSCs.”

Our response: We have made the suggested modification.

L501-502 – Change to “As we have previously mentioned, such elderly motivations could reside in culture or could be related to socializing needs of this age group.”

Our response: We have made the suggested modification.

L503-505 – Make sure NOT to capitalize cash and debit card so written as “cash” and “debit card”

Our response: We have made the suggested modification.

L506-508 – Change to “The density of values are represented most closely to the junction of both axes in our multivariate correspondence graphs. Cash or debit card payments show high densities which are close in terms of percentage.”

Our response: We have made the suggested modification.

L511-512 – Change to “…short food supply chains that are currently in an emergent phase in Romania.”

Our response: We have made the suggested modification.

L514-521 – These are a hanging solo sentence and a paragraph…merge into ONE paragraph.

Our response: We have made the suggested modification.

L491-521 The clusters of those ranging between 20-34 and 35-49 years old are approximately equally distributed on the horizontal and vertical lines, exhibiting a higher presence in Q1 and Q2. In this area it is about the high purchasing frequencies and partiality for cash payment. The cluster of those ranging between 50 and 65 years old is mainly distributed to the right side (Q2 and Q3). However, the difference from the distribution visible in Q1 and Q2 is not significant. Based on this distribution, we can reach the following conclusion: the inertia in the traditional buying behavior is rather higher within this age category, whilst there are also registered significant changes of the buying behavior on the short food supply chains. The cluster of the persons over 65 is mainly distributed in Q3 and shows a high resilience of the purchase of fresh vegetables directly from producers on short food supply chains. As we have previously mentioned, the motivations could reside in the cultural features or could be connected to the socializing needs of this age category. The most representative age categories in the case of the analyzed batch prefers to pay Cash and by Debit Card. Furthermore, those ranging 20-34 years old choose to pay by Debit Card and Cash, while those of 35-49 years old prefer to pay Cash and by Debit Card.

Regarding the density of the values (those represented most closely to the junction axes have the higher density), it can be noticed that cash payments or those made by debit card show high densities which are close in terms of percentage. In a Western society, this feature might look slightly concerning, given the strong digital transformation undergone by these cultures. In the case of Romania, the values recorded are still positive, and show an increase related to the degree of acceptance of digital transformation on short food supply chains (they are currently in an emergent phase in Romania). 

 Was changed to

Several aspects of age cluster data distribution warrant further consideration. The clusters of those ranging between 20-34 and 35-49 years old are approximately equally distributed on the horizontal and vertical lines, exhibiting a higher presence in Q1 and Q2. In this area it is about the high purchasing frequencies and partiality for cash payment. The cluster of those ranging between 50 and 65 years old is mainly distributed to the right side (Q2 and Q3). However, the difference from the distribution visible in Q1 and Q2 is not significant. This suggests the inertia of traditional buying behavior in short food supply chains (SFSCs) is rather higher for older consumers. The cluster of the persons over 65 is mainly distributed in Q3 and shows a high resilience of the purchase of fresh vegetables directly from producers on SFSCs. As we have previously mentioned, such elderly motivations could reside in culture or could be related to socializing needs of this age group. The most representative age categories in the case of the analyzed batch prefers to pay cash and by debit card. Furthermore, those ranging 20-34 years old choose to pay by debit card and cash, while those of 35-49 years old prefer to pay cash and by debit card.

The density of values are represented most closely to the junction of both axes in our multivariate correspondence graphs. Cash or debit card payments show high densities which are close in terms of percentage. In a Western society, this feature might look slightly concerning, given the strong digital transformation undergone by these cultures. In the case of Romania, the values recorded are still positive, and show an increase related to the degree of acceptance of digital transformation on short food supply chains that are currently in an emergent phase in Romania.. 

L517-518 – Change to “However, consumer avoid shopping at grocery stores, farmers’ markets and/or supermarkets which were often crowded during the pandemic.”

Our response: We have made the suggested modification.

L518-519 – Change to “Social distancing is respected and contact with unknown, possibly infected people is avoided.”

Our response: We have made the suggested modification.

L519-521 – This sentence does not make any sense…How can asymptomatic consumers have no contact with markets? This was not ascertained in your survey. Please delete this sentence or make clearer.

Our response: We have deleted the sentence.

L522-524 – Change this sentence to “To increase local food production and sales in Romania, small local producers need to adhere to shifting customer preferences and innovate their marketing strategies.”

Our response: We have made the suggested modification.

L529 – Change to “times of crises and beyond.”

Our response: We have made the suggested modification.

L530-549 – Write the last two paragraphs not in numerical order of recommendations but in more of a written format making connections between the points.

Our response: Given your suggestion, upon our latest revision aiming at delivering a more concise content. More precisely, we rewrote the paragraphs you mentioned and changed the style - narrative discourse.

The paragraphs:

Based on the analyses run in this study, a series of general recommendations stand out for the local producers: 1) adjusting the payment methods to the consumers’ demands by purchasing mobile POS systems; 2) developing their own brands and products at the same time with an integrated promotion (in analog and digital system); 3) implementing technologies for placing online orders by developing their own specialized websites and social media; 4) innovative marketing and planning of the distribution according to the customers’ demands and short food supply chains; 5) developing the association and cooperation forms for a better access to the market. 

At the same time, we strongly encourage the consumption of fresh vegetables directly purchased from producers, and the development of short food supply chains which bring the following benefits to the consumers: 1) the local products distributed by SFSC have commonly a superior nutritional value and a favorable impact upon the general condition and health; 2) employing SFSC nowadays and tomorrow attracts indirect economic benefits to consumers by retaining capital locally which, in turn, has a multiplying effect within the regional economy (maintaining and creating jobs, reinvested profit in productive activities, duties and taxes for the local revenue); 3) the acquisitions made within the short food supply chains contribute to environmental protection and hence improve the life quality, especially in urban areas; 4) the purchase of fresh vegetables directly from local producers on a regular basis tackles food waste; 5) by consuming local fresh vegetables, the consumer brings own contribution to the preservation of local tradition and identity (local gastronomy, local varieties of vegetables, rural culture, circular rural economy); 6) the direct delivery of fresh vegetables is time saving, shortening the time consumed for purchasing activities. 

Were changed to :

Based on the analyses run in this study, a series of general recommendations stand out for the local producers. Firstly, it could be advisable to adjust the payment methods to the consumers’ demands by purchasing mobile POS systems and, also develop their own brands and products with an integrated promotion (in analog and digital system). At the same time, and of equal importance, it is imperative to implement up-to-date technologies for placing online orders by developing their own specialized websites and social media. Additionally, innovative marketing and planning of the distribution should be made in accordance with the customers’ demands and short food supply chains. Last but not least, local producers could develop association and cooperation forms for a better access to the market.

For business development, certain limitations need to be taken into consideration. Firstly, supply with fresh vegetables directly from local producers cannot be achieved in Romania throughout the calendar year due to the seasonal nature of crops and reduced areas designated for greenhouses or poly tunnels [88,89]. The quantities and vegetable varieties locally produced and available on the market are reduced due to their zonal and seasonal nature, and, consequently, the demand cannot be exclusively covered by the local production [88]. Another key factor is the final cost of the product, as the price of the local vegetables sold through SFSCs is regularly higher than the prices of similar products in the hypermarket networks [90,91]. To address these issues and come up with viable solutions, the producers could follow the successful associative models from the Western Europe thus gaining more visibility and authority on the market. However, the reticence of Romanian small producers is a common denominator when it comes to associating and cooperating locally, including making a common brand, which cannot but become a  hindrance for penetrating the local market [92–94]. 

In Romania, the digital transformation of small producers can have a positive effect for the entire economy as the implementation of online commerce technologies and adjusting the payment methods for debit card and bank transfer can lead to increasing the taxation degree of the sales. However, the digital transformation is also influenced by certain local factors. For instance, to develop distribution channels, small producers need to invest in infrastructure. Nevertheless, their financial possibilities are rather limited and, they choose to invest in means of production at the expense of infrastructure for commercialization and marketing [95]. Furthermore, many small enterprises are in dire need of time, another impediment for digital transformation as they allot most of their activities to production and placing it on the market. Additionally, the scarce digital literacy of the producers is a barrier preventing and limiting the development of the innovative instruments of SFSC [96].

Concurrently, we strongly encourage the consumption of fresh vegetables directly purchased from producers, and the development of short food supply chains which bring a series of benefits to the consumers. Primarily, the local products distributed by SFSC have commonly a superior nutritional value and a favorable impact upon the general condition and health, and employing SFSC nowadays and tomorrow attracts indirect economic benefits to consumers by retaining capital locally which, in turn, has a multiplying effect within the regional economy (maintaining and creating jobs, reinvested profit in productive activities, duties and taxes for the local revenue). Additionally, the acquisitions made within the short food supply chains contribute to environmental protection and hence improve the life quality, especially in urban areas, not to mention the fact that the purchase of fresh vegetables directly from local producers on a regular basis tackles the issue of food waste. Also, by consuming local fresh vegetables, the consumer brings his / her own contribution to the preservation of local tradition and identity (local gastronomy, local varieties of vegetables, rural culture, circular rural economy). And, finally, the direct delivery of fresh vegetables is time saving, thus shortening the time spent on purchasing activities.

L551-553 – Change first sentence to “Our results confirms the hypothesis that the COVID-19 pandemic induced significant changes in consumer purchasing behavior of fresh vegetables.”

Our response: We have made the suggested modification.

L552 – Change to “…consumers are more determined to”

Our response: We have made the suggested modification and changed “….consumers are more and more determined…” as suggested above.

L557 – Change to “this system of buying from short food supply chains (SFSCs) after”

Our response: We have made the suggested modification.

L561 – Change to “…transformation of SFSCs.”

Our response: We have made the suggested modification.

L566-575 – These are two short paragraphs…merge into ONE paragraph.

Our response: We have made the suggested modification.

L568-569 – Change to “Thus SFSCs represent…”

Our response: We have made the suggested modification.

L574 – Change to “SFSCs” to be consistent

Our response: We have made the suggested modification.

L576-577 – Change to “…is tied to the current preliminary study of just the quarantined area of Suceava.”

Our response: We have made the suggested modification.

L579-580 – Change to “…the results of our study will add to the growing body of research on short food supply chains conducted nationwide in Romania as well as globally.”

Our response: We have  changed “ the results obtained by this study will provide new opportunities to novel lines of research conducted nationwide” as suggested above, to  “the results of our study will add to the growing body of research on short food supply chains conducted nationwide in Romania as well as globally “ .

Round 3

Reviewer 3 Report

This manuscript analyzes the impacts of the COVID-19 pandemic on consumer purchasing patterns for vegetables from local farmers in a county in Romania. I would like to thank the co-authors for taking the time to improve the quality of the manuscript and I recommend publication at this time following minor typographical corrections:

Specific Line Number of Manuscript edits (note that requested change of word(s) in quotations):

L30 – Change to “…related to current modern societies as they”

L32 – Change to “…and citizens’ mentalities as well.”

L33 – Change to “have been preserved over the medium term and long term)…”

L38 – Remove third and fourth commas

L40 – Remove comma after first word

L41 – Change to “…in recorded history…”

L46 – Change to “…on food security,”

L47 – Change to “and developing countries. Meanwhile, developed economies...”

L50-51 – Change to “…as the public focus of developed countries transitioned to issues such as healthy eating.”

L52 – Change to “During the Middle Ages,”

L57-58 – Change to “…movement of soldiers from WWI (as they came back home in 1918....”

L59-60 – Change to “…mobility of humans, mass socialization, as well as cultural, social, and tourism events…”

L66 – Change to “pandemics should be emphasized.”

L69 – Change to “During the census of 1930, Romania…”

L70 – Change to “…of material production…”

L71 – Change to “…agriculture, it is understandable that”

L73-74 – Change to “Additionally during that period, most of Romania’s population had a diet which…”

L74-75 – Change to “…inhabitants of Western Europe.”

L76-77 – Change to “…of the Spanish flu due to among other factors, a weakened immune system.”

L78 – Change to “However in Romania, the…”

L150 – Insert space between the two sentences

L197 – Change to “…food products / fresh food”

L226-229 – Change to “(3 – 1.2%). Most answers (87%) from all survey respondents (N=257) come from women. Other prevalent characteristics of respondents were being married, between 20 to 49 years old, having a Master’s degree and/or Bachelor’s degree, and living with two to four people in their household (Table 2).”

L236 – Change to “In these types of graphs, the four”

L383-384 – Delete the sentence not highlighted in yellow as it is redundant

L386 – Insert a space between the sentences

L390 – Change to “run small- or medium-sized farms…”

L392 – Change to “in affiliation with marketing platforms.”

L454-455 – Suggest deleting the “(121 respondents, 47%)” since it is not clear this corresponds to anything (e.g. 113 + 23 = 136 and not 121)

L457-459 – Change to “…(before, during and post crisis), the percentage of respondents who use cash payments were lower than for card payments or bank transfers (Figure 13).”

L459 – Change to “This demonstrates that during the COVID-19 crisis, the population became”

L472 – “thethe” is not a word

L472-474 – This is a stray sentence and should be integrated into the following full paragraph

L495-497 – Clarify how there can be correlation between monthly ordering and not ordering since March 16 as these two seem mutually exclusive

L508 – Change to “…fresh vegetables…”

L532 – Change to “There are numerous researchers…”

L533 – Change to “pandemics on the economy, especially on product distribution systems…”

L535 – Change to “of increases in consumption…”

L536-539 – Change to “human activities [78-80], especially for agriculture [81]. When it comes to the current pandemic, Carlsson-Szlezak [82,83] argue the three types of effects COVID-19 has had are on consumption, the market, and distribution chains.”

L539-541 – Change to “…should be redesigned to strengthen resilience in the future to address the complexity of contemporary society [84-87].”

L544 – Change to “…However, consumers avoid…”

L554-555 – Change to “…stand out for local producers.”

L555-556 – Change to “First, it is advisable for agricultural producers to adjust payment methods to consumers’ demands by purchasing mobile POS systems as well as to develop their own brands and products with”

L558 – Change to “it is imperative for producers to implement…”

L561-562 – Clarify this…are these actual paper forms? What are association and cooperation forms?

L563-564 – Change to “First, supply of fresh vegetables…”

L567 – Remove the second comma

L568 – Change to “exclusively covered by local production [88]. Another key factor is the final price of the product,”

L571 – Change to “…from Western Europe”

L572 – Change to “…authority in the market place.”

L574 – Change to “…a hindrance…” by removing extra space

L576 – Paragraph indent is too far over to the right

L576-578 – Clarify this since increasing sales tax can increase government revenue that can be spent to improve the economy…however, increasing sales tax increases the price of the product which reduces consumer surplus and prices certain consumers out of the market…it needs to be clarified how this has a positive on the entire economy

L578 – Change to “…to increasing sales taxes.”

L581 – Change to “financial possibilities are rather limited as they choose to invest…”

L584 – Change to “of their activities to production and bringing products to market.”

L585-586 – Change to “of agricultural producers is a barrier preventing and limiting the development of these innovative marketing instruments [96].”

L587 – Paragraph indent is too far over to the right

L588 – Change to “…short food supply chains (SFSCs) have…”

L589-592 – Change to “consumers. Typically local products distributed by SFSCs have superior nutritional value and favorable impacts on people’s general condition and health. Using SPSCs now and into the future can result in indirect economic benefits to consumers by retaining capital locally, which in turn, has a multiplying…”

L593 – Change to “…taxes for local revenue, etc.).”

L594 – Change to “acquisitions made within SFSCs contribute…”

L599-600 – Change to “Finally, the direct delivery of fresh vegetables saves time for consumers by reducing the time spend on purchasing food.”

L608 – Change to “chains (SFSCs) following the COVID-19 crisis.”

L618-620 – Change to “Thus SFSCs represent a viable solution to the pandemic, since in Romania’s current context, the reliability and safety of the conventional pattern of agricultural production has been brought into question.”

L627 – Change to “has been the main reasons for the geographic limit of our current research.”

Author Response

Dear Reviewer,

Thank you for your comments to improve our manuscript.

You can find below our answer to each of the suggested changes.

L30 – Change to “…related to current modern societies as they”

Our response:  We have changed “nowadays societies” to “ current nowadays societies”.

L32 – Change to “…and citizens’ mentalities as well.”

Our response: We have changed “ and mentalities as well” to ““…and citizens’ mentalities as well”

L33 – Change to “have been preserved over the medium term and long term)…”

Our response: We have changed “have been preserved on medium and long term” to “ have been preserved over the medium term and long term)…”

L38 – Remove third and fourth commas

Our response: We have done the suggested change.

L40 – Remove comma after first word

Our response: We have done the suggested change.

L41 – Change to “…in recorded history…”

Our response: we have deleted the word “the”.

L46 – Change to “…on food security,”

Our response: we have deleted the word “the”.

L47 – Change to “and developing countries. Meanwhile, developed economies...”

Our response: We have done the suggested change.

L50-51 – Change to “…as the public focus of developed countries transitioned to issues such as healthy eating.”

Our response: We have changed “as the public space of the developed countries was invaded by issues such as healthy eating” to ““…as the public focus of developed countries transitioned to issues such as healthy eating.”

L52 – Change to “During the Middle Ages,”

Our response: we have changed “During Middle Ages,” to “During the Middle Ages,”

L57-58 – Change to “…movement of soldiers from WWI (as they came back home in 1918....”

Our response: we have changed “due to the movement of the soldiers from the WW II (as they came home back in 1918” to “movement of soldiers from WWI (as they came back home in 1918....”

L59-60 – Change to “…mobility of humans, mass socialization, as well as cultural, social, and tourism events…”

Our response: We have done the suggested change.

L66 – Change to “pandemics should be emphasized.”

Our response: we have changed “pandemics should be underlined” to ““pandemics should be emphasized.”

L69 – Change to “During the census of 1930, Romania…”

Our response: we have changed “The census of 1930 Romania estimated” to “During the census of 1930, Romania…”.

L70 – Change to “…of material production…”

Our response: we have changed “…of the material production…” to “…of material production…”.

L71 – Change to “…agriculture, it is understandable that”

Our response: we have changed “it is understandably” to “it is understandable”.

L73-74 – Change to “Additionally during that period, most of Romania’s population had a diet which…”

Our response: we have changed “Additionally, in that period, most Romanian population had a food diet which” as suggested above.

L74-75 – Change to “…inhabitants of Western Europe.”

Our response: we have changed “inhabitants of the Western states” as suggested above.

L76-77 – Change to “…of the Spanish flu due to among other factors, a weakened immune system.”

Our response: we have changed “effects of the Spanish flu due to, among others, a weakened immune system.” as suggested above.

L78 – Change to “However in Romania, the…”

Our response: we have deleted the comma.

L150 – Insert space between the two sentences

Our response: we have done the suggested change.

L197 – Change to “…food products / fresh food”

Our response: we have resolved the issue.

L226-229 – Change to “(3 – 1.2%). Most answers (87%) from all survey respondents (N=257) come from women. Other prevalent characteristics of respondents were being married, between 20 to 49 years old, having a Master’s degree and/or Bachelor’s degree, and living with two to four people in their household (Table 2).”

Our response: we have done the suggested change.

L236 – Change to “In these types of graphs, the four”

Our response: we have done the suggested change.

L383-384 – Delete the sentence not highlighted in yellow as it is redundant

Our response: we have done the suggested change.

L386 – Insert a space between the sentences

Our response: we have resolved the issue.

L390 – Change to “run small- or medium-sized farms…”

Our response: We have changed “who run small or medium farms” to “run small- or medium-sized farms…”

L392 – Change to “in affiliation with marketing platforms.”

Our response: We have changed “in affiliation to marketing platforms to ““in affiliation with marketing platforms.”

L454-455 – Suggest deleting the “(121 respondents, 47%)” since it is not clear this corresponds to anything (e.g. 113 + 23 = 136 and not 121)

Our response: We have changed to “As shown in Table 7, almost half of the respondents (47%) have chosen cash payment, while bank transfer is the least preferred payment method”.

L457-459 – Change to “…(before, during and post crisis), the percentage of respondents who use cash payments were lower than for card payments or bank transfers (Figure 13).”

Our response: we have done the suggested change.

L459 – Change to “that during the COVID-19 crisis, the population became”

Our response: we have changed “……This demonstrates that during COVID-19 crisis, the population became” to “that during the COVID-19 crisis, the population became”

L472 – “thethe” is not a word

Our response: we have corrected error “thethe”to “the”.

L472-474 – This is a stray sentence and should be integrated into the following full paragraph

Our response: we have done the suggested change.

L495-497 – Clarify how there can be correlation between monthly ordering and not ordering since March 16 as these two seem mutually exclusive

  Our response: we have changed “In the quadrant 2 (Q2) of Figure 14 there are the strongest correlations among those who order fresh vegetables directly from producers on a monthly basis (once a month), those who have not ordered since March 16.”

To

Quadrant 2 (Q2) corresponds to those who order fresh vegetables directly from producers monthly, who have not ordered since March 16, are in the 50-64 age category, and prefer to pay by bank transfer.”

L508 – Change to “…fresh vegetables…”

Our response: we have done the suggested change.

L532 – Change to “There are numerous researchers…”

Our response: we have changed “There are numerous authors…” to “There are numerous researchers…”

L533 – Change to “pandemics on the economy, especially on product distribution systems…”

Our response: we have changed “pandemics on economy and, especially, on the product distribution systems” to “pandemics on the economy, especially on product distribution systems…”

L535 – Change to “of increases in consumption…”

Our response: we have changed “of the consumption increase” to “of increases in consumption…”

L536-539 – Change to “human activities [78-80], especially for agriculture [81]. When it comes to the current pandemic, Carlsson-Szlezak [82,83] argue the three types of effects COVID-19 has had are on consumption, the market, and distribution chains.”

Our response: we have done the suggested change

L539-541 – Change to “…should be redesigned to strengthen resilience in the future to address the complexity of contemporary society [84-87].”

Our response: we have done the suggested change

L544 – Change to “…However, consumers avoid…”

Our response: we have changed “…However, consumer avoid…” to “…However, consumers avoid…”

L554-555 – Change  to “…stand out for local producers”.

Our response: we have changed “…stand out for the local producers.” to “…stand out for local producers”.

L555-556 – Change to “First, it is advisable for agricultural producers to adjust payment methods to consumers’ demands by purchasing mobile POS systems as well as to develop their own brands and products with”

Our response: we have done the suggested change

L558 – Change to “it is imperative for producers to implement…”

Our response: we have changed “it is imperative to implement” to “it is imperative for producers to implement…”

L561-562 – Clarify this…are these actual paper forms? What are association and cooperation forms?

Our response: we have changed “Last but not least, local producers could develop association and cooperation forms for a better access to the market” to “Last but not least, local producers could associate in cooperative organizations for a better access to the market. “

L563-564 – Change to “First, supply of fresh vegetables…”

Our response: we have changed “Firstly, supply with fresh vegetables” to ““First, supply of fresh vegetables…”

L567 – Remove the second comma

Our response: we have done the suggested change.

L568 – Change to “exclusively covered by local production [88]. Another key factor is the final price of the product,”

Our response: we have changed “exclusively covered by the local production [88]. Another key factor is the final cost of the product” as suggested .

L571 – Change to “…from Western Europe”

Our response: we have changed “from the Western Europe” to ““…from Western Europe”

L572 – Change to “…authority in the market place.”

Our response: we have changed “authority on the market” as suggested.

L574 – Change to “…a hindrance…” by removing extra space

Our response: we have done the suggested change.

L576 – Paragraph indent is too far over to the right

Our response: we have resolved the issue.

L576-578 – Clarify this since increasing sales tax can increase government revenue that can be spent to improve the economy…however, increasing sales tax increases the price of the product which reduces consumer surplus and prices certain consumers out of the market…it needs to be clarified how this has a positive on the entire economy; L578 – Change to “…to increasing sales taxes.”

Our response: To eliminate the unclear statement we have changed to “In Romania, the digital transformation of small producers can have a positive effect for the entire economy”

L581 – Change to “financial possibilities are rather limited as they choose to invest…”

Our response: we have changed “financial possibilities are rather limited and, they choose to invest” as suggested above.

L584 – Change to “of their activities to production and bringing products to market.”

Our response: we have changed “of their activities to production and placing it on the market.” As suggested above.

L585-586 – Change to “of agricultural producers is a barrier preventing and limiting the development of these innovative marketing instruments [96].”

Our response: we have changed “of the producers is a barrier preventing and limiting the development of the innovative instruments of SFSC [96].” as suggested above.

L587 – Paragraph indent is too far over to the right

Our response: we have resolved the issue.

L588 – Change to “…short food supply chains (SFSCs) have…”

Our response: we have changed “short food supply chains” “…short food supply chains (SFSCs)…”

L589-592 – Change to “consumers. Typically local products distributed by SFSCs have superior nutritional value and favorable impacts on people’s general condition and health. Using SPSCs now and into the future can result in indirect economic benefits to consumers by retaining capital locally, which in turn, has a multiplying…”

Our response: we have done the suggested change

L593 – Change to “…taxes for local revenue, etc.).”

Our response: we have changed  “..taxes for local revenue) to “…taxes for local revenue, etc.).”

L594 – Change to “acquisitions made within SFSCs contribute…”

Our response: we have changed “acquisitions made within the short food supply chains” as suggested above

L599-600 – Change to “Finally, the direct delivery of fresh vegetables saves time for consumers by reducing the time spend on purchasing food.”

Our response: we have done the suggested change

L608 – Change to “chains (SFSCs) following the COVID-19 crisis.”

Our response: we have changed “chains (SFSCs) after COVID-19 crisis” as suggested above.

L618-620 – Change to “Thus SFSCs represent a viable solution to the pandemic, since in Romania’s current context, the reliability and safety of the conventional pattern of agricultural production has been brought into question.”

Our response: we have done the suggested change

L627 – Change to “has been the main reasons for the geographic limit of our current research.”

Our response: we have changed “has been the main reason of the fact that this whole research endeavor

This manuscript is a resubmission of an earlier submission. The following is a list of the peer review reports and author responses from that submission.

Round 1

Author Response

Dear reviewer,

We are genuinely grateful for taking the time and effort to write and develop your review. We took note of your suggestions and useful recommendations, like those made by all of the other reviewers involved. We can assure you that we`ve made major revisions to our draft paper, by adjusting our narrative while providing many additional clarifications as well as new information. We can also assure you that we have also addressed the issues concerning the figures, namely resolution.

Best regards,

Authors

Reviewer 2 Report

Summary: The authors´ stated aim is to establish the the changes occurred in the purchasing behavior of fresh vegetables straight from producers and by direct delivery during the COVID-19 pandemic in the quarantined area of Suceava. Despite this original objective, in the methods section they stated another aim: After data collecting and formatting, the analysis objectives have mainly aimed the identification of similarities and dissimilarities, categories of answers at the level of respondent cluster according to age and purchase behavior. This creates confusion along the manuscript. There is no statistical data analysis, which is a major weakness. Another problem is that the authors´ used a very colloquial language which greatly reduces the quality of the manuscript.  

Potential Areas for Improvement.

Line 27-29. The paragraph is too short, and no main idea can be drawn (the same for Line 31-41; L 164-165, L- 176-177, L 186-187, L 462-463).

Line 31-36. Reference.

Line 43-101. Reference. There are many sentences in the introduction part that don’t have any reference (L209-L221).

Figure 2, the image is not clear.

Line 297, consider using a less colloquial language: By consulting the literature published, it stands out the fact that, up to this date.

Line 300, the text says “some authors”, but there is only one reference.

Line 351-354. No relation with the previous paragraph.

Section 2.4.2. It is a broader concept, therefore it should precede SFSC.

Section 3.1. It is expected to have results, it is a literature review.

Line 432-436. A figure is required (localization map).

Figure 4 repeats the text of Lines 437-444. The figure is not clear what does the 10, 20, etc represent (this applies for the other figures).

Figure 5. Numbers in the figure are not clear enough.

Figure 6. Text should be before the figure.

Line 478. “We can safely draw the conclusion that we are dealing with a localization of the vegetable” colloquial language.

Line 511. the obvious emotional state of those put in quarantine. What do the authors mean by obvious emotional state, clarify.

Figure 12. The title seems more an statement that a title.

Line 614. SARS-Cov2 is used instantly from COVID-19. It should be best to retain only one.

Line 632. Due to the lack of statistical analysis, the affirmation remains more and assumption.

Line 804. This is a manuscript not a material.

Reference section. 60, 59 are not correctly cited. Most references are to not scientific articles. Those from scientific articles are not updated.

Author Response

Dear reviewer,

Thank you for the kind review of our paper. Your suggestions helped us to improve our manuscript. We have uploaded the improved version of the manuscript. Please find below our response to each of the comments.

Summary: The authors´ stated aim is to establish the the changes occurred in the purchasing behavior of fresh vegetables straight from producers and by direct delivery during the COVID-19 pandemic in the quarantined area of Suceava.

Despite this original objective, in the methods section they stated another aim: After data collecting and formatting, the analysis objectives have mainly aimed the identification of similarities and dissimilarities, categories of answers at the level of respondent cluster according to age and purchase behavior.

This creates confusion along the manuscript. There is no statistical data analysis, which is a major weakness. Another problem is that the authors´ used a very colloquial language which greatly reduces the quality of the manuscript.

Potential Areas for Improvement.

Line 27-29. The paragraph is too short, and no main idea can be drawn (the same for Line 31-41; Line 31-36, L 164-165, L- 176-177, L 186-187, L 462-463).

Line 43-101. Reference. There are many sentences in the introduction part that don’t have any reference (L209-L221).

Our response: Some of the paragraphs have been deleted. Others have been adjusted so as to deliver a clearer and more concise message. Last but not least, additional useful references were provided.

Figure 2, the image is not clear.

Our response: We decided to delete Figure 2, given that it was not clear enough.

Line 297, consider using a less colloquial language: By consulting the literature published, it stands out the fact that, up to this date.

Our response: We took note of your suggestion and we`ve revised the text so as to avoid colloquial language.

Line 300, the text says “some authors”, but there is only one reference.

Our response: We took action on this matter and we`ve provided additional references.

Line 351-354. No relation with the previous paragraph.

Our response: Following a more attentive reading, we`ve decided to make due without the aforementioned paragraph.

Section 2.4.2. It is a broader concept, therefore it should precede SFSC.

Our response: We took note of your suggestion and as a result we`ve reconsidered the structure of the text and place of the section. That part now precedes the SFSC part.

Section 3.1. It is expected to have results, it is a literature review.

Our response: The issue has been addressed. The paragraph was moved to Chapter 2 so as to further develop the conceptual framework.

Line 432-436. A figure is required (localization map).

Our response: The area subjected to study (Suceava Containment Area) can be observed in Figure 3.

Figure 4 repeats the text of Lines 437-444.

Our response: Following a more attentive reading we`ve decided to eliminate the figure altogether.

The figure is not clear what does the 10, 20, etc represent (this applies for the other figures).

Our response: We`re grateful for your kind feedback. We took note of your suggestion and we decided to revise all of the other figures so as to address this particular issue.

Figure 5. Numbers in the figure are not clear enough.

Our response: We can confirm that the issue has been addressed.

Figure 6. Text should be before the figure.

Our response: We`ve noticed this particular aspect and we`ve taken action on the matter.

Line 478. “We can safely draw the conclusion that we are dealing with a localization of the vegetable” colloquial language.

Our response: We consider your suggestion as justified. We agree and we`ve decided to make do without colloquial language.

Line 511. the obvious emotional state of those put in quarantine. What do the authors mean by obvious emotional state, clarify.

Our response: We took note of your suggestion. We decided to rephrase so as to clarify the issue we were trying to describe.

Figure 12. The title seems more an statement that a title.

Our response: The title came about as a result of the three questions, implicitly their length, part of our questionnaire. That is why it may also lead one into thinking of it as a statement.

Line 614. SARS-Cov2 is used instantly from COVID-19. It should be best to retain only one.

Our response: We took note of your suggestion and we`ve reconsidered our spelling.

Line 632. Due to the lack of statistical analysis, the affirmation remains more and assumption.

Our response: Given the high number of respondents who stated that they rather choose the products and quantities on their own is indicative of conclusive information, beyond any reasonable assumption. Nonetheless, following a more attentive reading we reconsidered our narrative discourse so as to provide one with better insight.

Line 804. This is a manuscript not a material.

Our response: We took note of the matter indicated and we`ve acted accordingly.

Reference section. 60, 59 are not correctly cited.

Our response: We can confirm that these issues were addressed. To be more precise, we`ve deleted the paragraphs indicated aiming at being more concise. problem was solved by removing the paragraphs, aiming at simplifying the text.

Most references are to not scientific articles. Those from scientific articles are not updated.

Our response: We took note of your suggestion and as a result we`ve taken action in this regard so as to follow your recommendation.

Best regards,

Authors

Reviewer 3 Report

The paper is interestig and is considering an actual problem.

The literature reviewe is very poor, I recommend to read some papers of this Journal and  to improve the references.

I think it's necessary to add the Hypothesis or the assumpion to improve the readibility of the paper

the limit of the reserach are missed, but i think they are very important to evalutate the results

In my opinion further analysis could be made, because the analysis of the datamis very simple.

Author Response

Dear reviewer,

Thank you for the kind review of our paper. Your suggestions helped us to improve our manuscript. We have uploaded the improved version of the manuscript. Please find below our response to each of the comments.

The paper is interesting and is considering an actual problem.

Our response: We would like to thank you for your appreciation.

The literature review is very poor, I recommend to read some papers of this Journal and to improve the references.

Our response: The issue has been addressed and ultimately resolved, as recommended.

I think it's necessary to add the Hypothesis or the assumption to improve the readability of the paper

Our response: According to the reviewer's suggestions, we`ve specified the working hypothesis as well as the assumptions in Chapter 2.

The limit of the reserach are missed, but i think they are very important to evalutate the results

Our response: The issue was addressed by specifying the limitations of research and potential development opportunities.

In my opinion further analysis could be made, because the analysis of the data is very simple.

Our response: Several new analyses, clarifications and interpretations were made and are now inserted in Chapters 2,3 and 4.

Best regards,

Authors

Reviewer 4 Report

The paper presented deals with a consumer analysis relating to a territorial area subject to lockdown during the COVID-19 2020 pandemic.
The subject of the market analysis is related to the purchase intentions of consumers of fresh vegetable products from local producers. This analysis was conducted by administering an online questionnaire to local consumers.

In the introduction, the paper reports the events that led to the imposition of lockdown in the territorial area under study (lines 43-101). This report appears excessively long and, while agreeing for its presence to understand the territorial context, it has to be reformulated in content and reduced in size. In addition, the entire paragraph 1.3 should be merged with the previous introductory description, avoiding the use of the detailed list of the regulatory governmental provisions that do not add useful information for the reader to understand the topic. Finally, the graph shown in figure 2, showing the epidemiological diffusion in the areas under study, should be commented on further and its source mentioned directly.

In the materials and methods section, there is a generic reference to the identification of similarities and dissimilarities with reference to age classes and purchasing behavior. The statistical methodology used (software package R) makes no mention of which multivariate method of data extraction was used, not identifiable even by using the bibliographic references provided.

The entire data discussion is based on reading the graphs obtained from the aforementioned (but not expressed) statistical methodologies, and the analysis of the results shows a mere representation by classes of the results of the questionnaire administered.
Furthermore, some of the graphs appear difficult to understand because of the choice of the type of graphic display proposed (see the spider plot in fig. 11).

The comment on the multivariate graph in fig. 13 (defined by the authors “multivariable”) is not adequately commented and not clearly identifiable labels further complicate the understanding of the results.

The given conclusions, therefore, do not appear to be adequately based on the results inferable from the information reported in the proposed graphs. It is thus necessary to review its content after the results have been reworked.

It can be therefore concluded that the article requires a major revision to be taken into consideration for the journal, focusing corrective interventions on the rigorous exposure of the statistical methodologies used for data analysis and a rewriting of the results and their discussion.

The English form of the text appears to be sufficiently correct, except for a few spelling inaccuracies in text and figures. Therefore, the text requires critical reading, once the recommended changes have been made, by an English native speaker.

Author Response

Dear reviewer,

Thank you for the kind review of our paper. Your suggestions helped us to improve our manuscript. We have uploaded the improved version of the manuscript. Please find below our response to each of the comments.

The paper presented deals with a consumer analysis relating to a territorial area subject to lockdown during the COVID-19 2020 pandemic. The subject of the market analysis is related to the purchase intentions of consumers of fresh vegetable products from local producers. This analysis was conducted by administering an online questionnaire to local consumers.

In the introduction, the paper reports the events that led to the imposition of lockdown in the territorial area under study (lines 43-101). This report appears excessively long and, while agreeing for its presence to understand the territorial context, it has to be reformulated in content and reduced in size.

Our response: We took note of your suggestion and following a more attentive reading we`ve decided to do several significant changes concerning our discourse and approach. We`d like to thank you for your useful insight.

In addition, the entire paragraph 1.3 should be merged with the previous introductory description, avoiding the use of the detailed list of the regulatory governmental provisions that do not add useful information for the reader to understand the topic.

Our response: After having looked over once more over the section that you kindly indicated we`ve reconsidered our tackling of the topic. We fully agree with you and as a result we`ve addressed the issue according to your suggestions and indications.

Finally, the graph shown in figure 2, showing the epidemiological diffusion in the areas under study, should be commented on further and its source mentioned directly.

Our response: Given your opinion on the matter we`ve decided to delete the graph altogether.

In the materials and methods section, there is a generic reference to the identification of similarities and dissimilarities with reference to age classes and purchasing behavior.

Our response: As a result of our review of the aforementioned we`ve reconsidered our view on the structure of the paper, materials and methods section included. As a result, we then decided to significantly extend our methodological approach, aiming at providing further clarifications on this matter.

The statistical methodology used (software package R) makes no mention of which multivariate method of data extraction was used, not identifiable even by using the bibliographic references provided.

Our response: Given the issue indicated, we`ve taken action so as to improve our shortcomings. Undoubtedly justified, we decided to better describe our methodological approach. More precisely, we addressed and paid particular attention the Multiple Correspondence Analysis part.

The entire data discussion is based on reading the graphs obtained from the aforementioned (but not expressed) statistical methodologies, and the analysis of the results shows a mere representation by classes of the results of the questionnaire administered.

Our response: Your suggestions on the matter were addressed and solved in the section dedicated to methods and methodology, which was further extended, as previously stated.

Furthermore, some of the graphs appear difficult to understand because of the choice of the type of graphic display proposed (see the spider plot in fig. 11).

Our response: We agree that due to some resolution related issues some graphs were not that clear. Accordingly, we rendered the graphs once more, as advised.

The comment on the multivariate graph in fig. 13 (defined by the authors “multivariable”) is not adequately commented and not clearly identifiable labels further complicate the understanding of the results.

Our response: Given your aforementioned, we considered adding more insight on the matter so as to improve our comments and arguments. We aimed at providing a better explanation and description.

The given conclusions, therefore, do not appear to be adequately based on the results inferable from the information reported in the proposed graphs. It is thus necessary to review its content after the results have been reworked.

Our response: We took into account your view on the matter and we proceeded accordingly. After having had completed an extensive review of the text several times we decided to provide a new version the chapter dedicated to the Conclusions section, as recommended.

It can be therefore concluded that the article requires a major revision to be taken into consideration for the journal, focusing corrective interventions on the rigorous exposure of the statistical methodologies used for data analysis and a rewriting of the results and their discussion.

Our response: We took note of your opinion and we can assure you that we acted accordingly. Our analyses and results were either further developed or completely revised altogether, as indicated, focusing on corrective interventions.

The English form of the text appears to be sufficiently correct, except for a few spelling inaccuracies in text and figures. Therefore, the text requires critical reading, once the recommended changes have been made, by an English native speaker.

Our response: We fully agree with you that our text needed an overall review aiming at eliminating any spelling and grammar related inaccuracies. We paid more attention to this rather important issue that is English accuracy. We`d really like to thank you for providing us with the necessary and useful suggestions and recommendations which have proved to be very useful to us and our draft paper.

Best regards,

Authors

Round 2

Reviewer 1 Report

You have improved this paper after the major revision, but I still have some confusion after reading the article. There are in the following aspects.

Introduction

This article mainly focuses on the consumers' purchasing behaviours of fresh vegetables, and it is unnecessary to devote too much to introduce the COVID-19 (line 27 to line 385). The epidemic situation should indeed be presented in this section, but more importantly, the research question needs to be introduced as soon as possible. You only show your research in line 388 to line 406.

Theoretical Contribution

Your research mainly focuses on the consumers' purchasing behaviours, so maybe you should use some consumer-related theories to explain consumer behaviour. And you'd better point out the difference between your research and previous research, as well as your contribution to the field of study.

Materials and Methods

Your materials and methods part contains too much information, such as hypothesis and results and conclusion. But they are separate parts.

I am confused why you hypothesized in this part. And the hypothesis should be derived, and it is logical. But in this part, you should focus on the research design and methods.

It is necessary to think about whether the demographic profile from the Quarantined Area should be put in this part or not.

What's more, the analysis and interpretation are the results of the survey. It is not appropriate to put here.

Analyses, Interpretations and Results

I am confused about why you have two analysis and interpretation part (part 2.4 and part 3). This section usually appears only once in an article.

Other Comments

As comments in the last edition, the figure is so misty that no one can see the information in the picture.

Reviewer 3 Report

The paper could be and interesting paper but there are some gaps. 

I caonnot find he hypothesis in the paper. You wrote: "The working hypothesis based on the exceptional situation triggered by the quarantine of 425 Suceava city and its peri-urban localities that, in turn, produced a series of behavioral changes 426 observed in the case of the consumers of fresh vegetables". this is not an hypothesis. I think it's necessary to add the Hypothesis or the assumpion to improve the readibility of the paper.  

The limit of the reserach must be connected with the hypothesis and they are very important to evaluate the results. Normally the limitation are in the conclusions.

In my opinion statistical analysis i necessary to improve the paper. 

Reviewer 4 Report

The authors took into account the general indications provided in the first review and proceeded to a thorough rewriting of the paper.

The paper, however, still presents serious weaknesses, such as a residual text redundancy and excess of information in the introduction and above all from a methodological point of view in using appropriate statistical methods for processing the survey information.

In materials and methods, paragraph 2.5 related to the conceptual context should be moved to the beginning, or identified as a separate paragraph.

In the results, as already reported in the previous review, the spider plot graph used in fig.10 is still not clearly understandable to the reader, and we recommend replacing it with another form of representation or make it uniform with the other bar-type graphs.

Throughout the results, there is no descriptive statistical validation on the individual answers provided. Furthermore, the multivariable analysis reported in figure 12, indicated as "multivariable correlations" (and probably referring to the "multiple correspondence analysis" routine present in the statistical package R) the authors do not indicate why this analysis was chosen over the other multivariate analyses. Also, the graph does not allow an immediate verification of the statements reported in the paper.

However, the results of the survey, as also stated by the authors, could be probably overthrown by a subsequent analysis conducted on the same sample in conditions of absence of lockdown. Therefore, the study would have greater significance if it were supplemented by a post-lockdown analysis of consumer behavior.

In conclusion, even if the authors have made the purpose of the research more understandable after their critical review, the whole paper is still difficult to read and still presents the need for additions (at a statistical level) to validly support the reported conclusions.

The English form is substantially correct.